# Two-Hop Cooperative Caching and UAVs Deployment Based on Potential Game

Yuan Bian [1],*, Jianbo Hu [1], Shuo Wang [2,3], Yukai Hao [4], Wenjie Liu [1] and Chaoqi Fu [1]

1   School of Equipment Management and UAV Engineering, Air Force Engineering University, Xi'an 710043, China
2   State Key Laboratory of Astronautic Dynamics, Xi'an Satellite Control Center, Xi'an 710043, China
3   State Key Laboratory of Integrated Services Networks, Xidian University, Xi'an 710071, China
4   Xi'an Institute of Aviation Computing Technology, Xi'an 710043, China
*   Correspondence: bianyuanlara@163.com

**Abstract:** This paper explores the joint cache placement and 3D deployment of Unmanned Aerial Vehicle (UAV) groups, utilizing potential game theory and a two-hop UAV cooperative caching mechanism, which could create a tradeoff between latency and coverage. The proposed scheme consists of three parts: first, the initial 2D location of UAV groups is determined through K-means, with the optimal altitude based on the UAV coverage radius. Second, to balance the transmission delay and coverage, the MOS (Mean Opinion Score) and coverage are designed to evaluate the performance of UAV-assisted networks. Then, the potential game is modeled, which transfers the optimization problem into the maximization of the whole network utility. The locally coupling effect resulting from action changes among UAVs is considered in the design of the potential game utility function. Moreover, a log-linear learning scheme is applied to solve the problem. Finally, the simulation results verify the superiority of the proposed scheme in terms of the achievable transmission delay and coverage performance compared with two other tested schemes. The coverage ratio is close to 100% when the UAV number is 25, and the user number is 150; in addition, this game outperforms the benchmarks when it comes to maximizing MOS of users.

**Keywords:** 3D UAV deployment; proactive cooperative cache; potential game; two-hop neighbors

## 1. Introduction

With the rapid development of 5G communication technology and the ever-increasing data demands from mobile devices, the current number and placement of ground-base stations are inadequate to meet the exploding demand [1]. Particularly in challenging environments, such as those that are hostile or unknown, this dearth not only creates economic pressure but also results in empty loads during non-off-peak traffic periods. To address this, UAVs equipped with wireless transceivers offer high mobility, deployment flexibility, and low cost, enabling them to serve as aerial base stations that can provide data services for some areas, including major events and conference activities, regardless of geography [2].

As a further advancement, a cache-enabled UAV has become an effective solution by caching popular contents in mobile edge networks, which alleviates the access load of ground-base stations, reduces the transmission delay, and improves the quality of the user experience. Thus, especially in this era of data explosion, users could fetch the requested contents from a cache-enabled UAV with less waiting time and fewer network congestion issues than with current technology. Therefore, a cache-enabled UAV-assisted network is considered a promising technology for dealing with the relevant challenges [3,4].

## 2. Related Works

In recent years, significant attention has focused on cache-enabled UAV-assisted networks as a means of ensuring quality of service [5]. Extensive research has been conducted on various aspects, including the channel-state information and performance, such as the communication model, bit error rate (BER), and channel capacity. In [6], a statistical propagation model was proposed to predict the air-to-ground path loss between a low-altitude platform and a terrestrial terminal, which characterized the air-to-ground path into two distinct path-loss profiles. Furthermore, ref. [7] provided statistical models for air-to-ground radio channels in dense urban environments, demonstrating that airborne platforms can act as relaying nodes to extend the range and improve connectivity between terrestrial ad hoc terminals. Additionally, refs. [8,9] presented an elaborate analysis of mixed RF/FSO systems, providing an integrated investigation of UAV-assisted wireless communication systems.

Moreover, critical issues of UAV-assisted networks have been classified into three groups: 3D deployment, resource allocation, and UAV trajectory.

With regard to 3D deployment, there are various performance metrics, such as coverage, connectivity, energy, and throughput, which have been used as objectives for optimizing UAV deployment. For example, in [10], the authors addressed the issue of ground-target coverage using UAVs while ensuring connectivity. In [11], the focus was on maximizing the coverage region of a single UAV by optimizing its vertical and horizontal dimensions and minimizing transmit power. Furthermore, ref. [12] proposed an algorithm to minimize task completion time in UAV-enabled MEC systems based on SCA. In [13], potential game theory was used to control the quasi-stationary deployment of UAVs and maximize the downlink wireless coverage of a UAV swarm in an unknown mission area. Additionally, ref. [14] considered location, power control, and connectivity to address the area coverage problem. In [15], the deployment scheme of UAVs was adjusted using Virtual Force Field theory to maximize the total network throughput based on statistical user position information.

With respect to joint 3D deployment and resource allocation, ref. [16] addressed the joint placement of UAVs and their association with users to maximize the network sum-rate under bandwidth limitation and quality of service constraints. In contrast, ref. [17] formulated the problem of the long-term caching placement and the optimization of resource allocation to minimize content delivery delay as a Markov decision process solved by Q-learning. Meanwhile, in [18], the focus was on investigating jointly UAV deployment and power allocation in a UAV-assisted MIMO-NOMA WCN (wireless caching network) to minimize user delay.

With regard to trajectory optimization, ref. [19] studied the optimization of a single UAV's trajectory based on a new design paradigm of communication throughput and energy consumption. In [20], the optimization of deployment and movement for multiple UAVs was studied while considering several ground terminals (GTs) communicating with the UAVs using variable transmission power and a fixed data rate. The corresponding trajectory optimization algorithm introduced was shown to guarantee a convergent Lagrangian. Furthermore, ref. [21] considered a practical 3D urban environment with imperfect CSI and designed the UAV's trajectory to minimize the completion time of data collection while adhering to practical throughput and flight movement constraints.

Previous studies have focused on optimizing both UAV deployment and content placement for cache placement to enhance overall network performance. In [22], an optimal configuration for content-aware UAV-assisted network content caching and location services was suggested, taking into account the correlation between users and the surroundings. This approach considered a trade-off between the user's service probability and transmission overhead when sharing cached content among one-hop neighbors. However, the system model consists of a substantial central UAV and several service UAVs, with the former being accountable for delivering the cached content to designated UAVs for collaborative purposes. The research paper failed to account for the flight duration of the

central UAV and the communication cost of real-time information exchange between each service UAV and central UAV.

The focus of [23] pertains to a scenario where a single UAV handles random and asynchronous content requests for ground nodes. In contrast to [22], the files were cached in specific ground nodes during the initial phase of each operation cycle and shared via device-to-device (D2D) communication. The optimization problem was aimed to minimize the weighted sum of file caching and retrieval costs by jointly designing the file caching strategy, UAV flight trajectory, and transmission scheduling. The extension of the proposed scheme to encompass multiple UAVs and the storage of files in both UAVs and ground terminals is a task left for future exploration.

In [24], the authors presented a joint optimization problem aimed at maximizing the quality of experience (QoE) of users by mean opinion score (MOS) through the deployment of UAVs, caching placement, and user association. The solution was decomposed into three sub-problems, namely, swap-matching-based UAV deployment, greedy-based caching placement, and Lagrange dual-based user association. It is worth noting that the paper provided a list of candidate UAV deployment locations.

The objective of the proposed study, as outlined in reference [25], was to enhance the QoE evaluated through the content delay index (CDI) while considering the latency in delivering content to mobile users on the ground. To achieve this, the optimization process problem was decomposed into three stages following the procedure of optimizing the 2D position, height, and proactive content caching. It could be found that UAVs serve autonomously for ground users, with no collaborative caching mechanism in place.

In another study, ref. [26] proposed a three-layer cache architecture for UAVs that enabled hierarchical adaptation to the dynamic changes of users and UAVs. By utilizing a user-adaptive UAV trajectory model in the UAV-based MEC layer, as well as a UAV-adaptive cache model in the cognitive center layer, both the transmission efficiency and hit rate of the system were improved. Additionally, ref. [27] utilized mean-field game theory to optimize the placement of content in UAVs by modeling user social attributes as spatial and temporal attributes.

Regarding cache placement, previous works [24–27] have primarily focused on two aspects: on the one hand, some works investigated the impact of users' attributes on content placement and utilized mathematical models to demonstrate the relationship between users and the deployment of UAVs using indicators such as transmission delay, MOS, and CDI. On the other hand, other works concentrated on trajectory [28] or transmission-scheduling [29] techniques with a focus on transmission power and content caching. While most of the existing research concentrated on individual UAV caching strategies or cooperative, complementary content transmission of one-hop [30], very few studies have considered cooperative transmission among UAVs of two-hop. Such an approach could significantly reduce the number of times UAVs need to access the MBS to obtain un-cached files. Additionally, the utility function design as in reference [24] did not consider the locally coupling effect between the UAV and its two-hop neighborhood, and any action taken by the UAV could lead to changes in the utility values of its neighborhhood. Consequently, designing a joint, efficient, proactive two-hop cooperative-content caching and UAV-deployment scheme while also taking into account the interdependence among UAVs poses a significant challenge.

Motivated by the above observations, in this paper, we propose a cache-able UAV-assisted network system where a cooperative transmission strategy and a spatial adaptive UAV deployment are jointly adopted to reduce transmission delay and improve coverage. Table 1 illustrates the contrasting analysis of the suggested methodology against other relevant sources. The contributions of this paper are summarized as follows:

1.  First, UAV cache placement and deployment are combined to optimize system efficiency considering communication delay and coverage. To enhance the utility of the network, a two-hop UAV cooperative-caching mechanism is proposed.

2. Second, we aim to formulate the problem of joint cooperative caching and 2D placement optimization as a strict potential game. To design the utility function of the potential game, we consider the locally coupling effect resulting from action changes among UAVs. The problem is transferred to maximize the whole network utility, which is defined by jointly considering MOS and coverage.

3. Third, the log-linear learning scheme is proposed to arrive at the solution of the potential game.

**Table 1.** Comparison of literatures.

| Reference | Cooperative Cache Strategy | Cache in UAV | Metrics | Approach | Candidate Location of UAVs | Notes |
|---|---|---|---|---|---|---|
| [22] | √ One-hop collaborative | √ | User's service and probability transmission overhead | Potential game | unknown | The duration of the central UAV's flight and the costs of communication for real-time information exchange between each service UAV and the central UAV have not been factored in. |
| [23] | × | × | Weighted sum of file caching and retrieval costs | Swap matching Greedy algorithm Lagrange dual | trajectory | Only consider the single UAV which does not provide service for users. |
| [24] | × | √ | MOS | Decomposition the optimization problem | known | The coverage has not been considered. |
| [25] | × | √ | CDI | Decomposition the optimization problem | unknown | The joint consideration of cache content and UAV horizontal position has not been taken into account. |
| [26] | × | √ | Transmission efficiency and hit rate | Trajectory and cache model | unknown | Collaborative caching of UAVs could be considered in the future. |
| [27] | × | √ | User social attributes | Mean-field game | unknown | The mean-field game is mainly for a large number of players |
| [28] | × | √ | Throughput | Block alternating descent and successive convex approximation | trajectory | The joint consideration of cache content and UAV horizontal position has not been taken into account. |
| [29] | × | √ | Access delay and cache-hit delay | Decomposition the optimization problem | trajectory | Collaborative cache mechanism could be considered in the future. |
| [30] | √ One-hop collaborative | √ | Transmission reliability and transmission energy consumption | Coalition Formation Game | unknown | Locally coupling effect between UAVs is not considered. |
| proposed | two-hop collaborative | √ | Modified MOS and. Coverage | Decomposition the optimization problem. Potential game | unknown | Please refer to the Conclusions and Future Work section for further details. |

The subsequent sections of this paper are structured as follows: Section 3 presents a detailed description of the system model, while Section 4 introduces a joint proactive cooperative-content caching scheme and UAV deployment strategy. The proposed approach leverages the log-linear caching algorithm to effectively achieve the desired outcomes. Simulation experiments and discussion are carried out in Section 5, and the conclusion is drawn in Section 6.

### 3. System Model and Problem Definitions

In this section, we begin by presenting the system model (Section 3.1) in mathematical terms, followed by an explanation of the transmission model (Section 3.2), network model (Section 3.3), and content cache model (Section 3.4). As caching storage on a single UAV is limited, we propose a mechanism in Section 3.5 that utilizes cooperative content caching to minimize transmission delay. In Section 3.6, we formulate a weighted function with associated constraints to evaluate critical attributes that impact the system performance.

#### 3.1. System Model

As depicted in Figure 1, the network model comprises a macro base station (MBS) that connects to the core network and several Unmanned Aerial Vehicles (UAVs) that act as edge servers while hovering in the air. All UAVs are connected with the MBS via wireless links. Additionally, much user equipment (UE) is present on the ground as consumers accept the services from the UAVs. This network model adequately portrays a UAV-assisted communication system that can meet increasing data traffic needs.

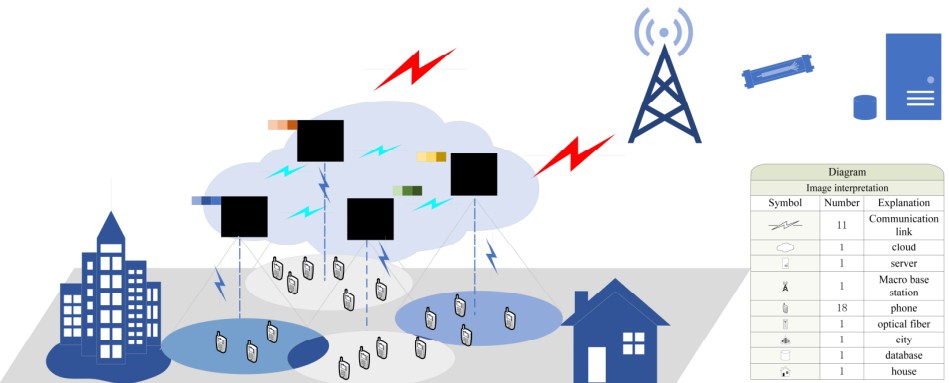

**Figure 1.** The total utility of the system with varying numbers of ground users.

Assuming the presence of UAV-BSs, denoted by the set $\mathcal{N}$, $\mathcal{N} \triangleq \{1, 2, 3, \cdots N\}$, flying at the same fixed height $h$. The 2D position of UAV-BS $n$ is denoted by $l_n$, where $l_n = (x_n, y_n) \in \mathbb{R}^2$. There exist $I$ static UEs with randomly assigned locations, and the location of UE is denoted by $l_i = (x_i, y_i) \in \mathbb{R}^2$. Assumes that the cache capacity of each UAV is H bits, and the UAVs receive the un-cached files from the Macro Base Station (MBS) through a wireless link while also actively caching multiple copies of popular content during off-peak hours. The variables of the system are shown in Table 2.

**Table 2.** Nomenclature of Variables.

| Symbol | Description |
| --- | --- |
| $N$ | Number of UAVs |
| $I$ | Number of users |
| $h$ | Fixed UAV height |
| $H$ | UAV cache capacity |
| $J_n$ | Two-hop range neighbors of UAV $n$ |
| $J_{1-n}$ | One-hop neighbors of UAV $n$ |

**Table 2.** *Cont.*

| Symbol | Description |
|---|---|
| $J_{2-n}$ | Two-hop neighbors of UAV $n$ |
| $M$ | Size of each file |
| $\tau_{f,n}$ | Indicator of whether UAV $n$ caches content $f$ |
| $\sigma_{i,n}$ | Indicator of the link between UAV and user |
| $d_{n,i}$, $d_{n,m}$, $d_{n,MBS}$ | Distance between UAV and user, UAV and UAV, MBS and UAV |
| $\theta_{i,n}$, $\theta_{n,MBS}$ | Elevation angle of UAV-to-User, MBS-to-UAV |
| $P_{Los}$, $P_{NLos}$ | The probability of LoS link and NLoS link connection between UAV $n$ and the ground user UE $i$ |
| $P_{Los}^{n,MBS}$, $P_{Los}^{n,MBS}$ | The probability of LoS link and NLoS link connection between MBS and UAV $n$ |
| $PL_{n,i}$, $PL_{n,m}^{Los}$, $PL_{n,MBS}$ | Pathloss of UAV-to-User link, UAV-to-UAV link, MBS-to-UAV link |
| $SNR_{n,i}$, $SNR_{n,m}$, $SNR_{n,MBS}$ | SNR of UAV-to-User link, UAV-to-UAV link, MBS-to-UAV link |
| $C_{n,i}$, $C_{n,m}$, $C_{n,MBS}$ | Transmission rate of UAV-to-User link, UAV-to-UAV link, MBS-to-UAV link |
| $D_{i,n,f}$ | Transmission delay |
| $P_{i,f}$ | The probability that user $i$ requests for file $f$. |

### 3.2. Network Model

Given the low transmission rates of each UAV independently utilizing the backhaul link to receive a file from the MBS, a collaborative caching approach using a two-hop mechanism has been proposed to enhance the data service performance. The associated UAV can request the un-cached files from its two-hop range neighbors rather than the MBS. These neighbor nodes constitute the set $J_n = J_{1-n} \cup J_{2-n}$ denoted below:

$$J_{1-n} = \{ UAV_m | \|l_m - l_n\| \le l_{th}, \ m, n \in \{1, 2, 3, \cdots N\}\}, m \ne n \tag{1}$$

$$J_{2-n} = \{ UAV_{m'} | \|l_{m'} - l_m\| \le l_{th}, \ m, m' \in \{1, 2, 3, \cdots N\}\}, m' \ne m \ or \ n \tag{2}$$

where $l_{th}$ is the maximum communication distance between two UAVs. Additionally, to prevent serious interference and potential collision during flight, the distance between any two nodes must meet the following condition:

$$\|l_m - l_n\| \ge d_{min}, m, n \in \{1, 2, 3, \cdots N\}, m \ne n \tag{3}$$

where $d_{min}$ is the minimum safe distance between any two nodes.

### 3.3. Transmission Model

The transmission link of UAV-to-UE, UAV-to-UAV, MBS-to-UAV is modeled by 3GPP [31].

I.    UAV-to-UE link

As discussed in [6,32,33], when considering the link between the UAV and users, the ground receiver obtains three types of signals: line-of-sight (LOS), strong reflected signals (NLOS), and multiple reflected components that can cause multi-path fading. These signal groups can be examined individually, each with varying probabilities of occurrence. Typically, as discussed in [34], the received signal is assumed to belong to only one of these groups. The probability of occurrence for each group is dependent on the environment,

building density and height, and the elevation angle. LOS and strong NLOS components are more likely to occur than fading, as explained in [34,35], rendering the impact of small-scale fading negligible. A common approach to modeling the air-to-ground propagation channel involves analyzing the LOS and NLOS components separately and considering their respective probabilities of occurrence, which are denoted by:

$$PL_{n,i}^{Los} = 20log(4\pi f d_{n,i}/c) + \eta_{Los} \tag{4}$$

$$PL_{n,i}^{NLos} = 20log(4\pi f d_{n,i}/c) + \eta_{NLos} \tag{5}$$

where $d_{n,i}$ is the distance between UAV $n$ and ground user UE $i$; $f$ is the carrier frequency; $c$ represents the speed of light. $\eta_{Los}$ and $\eta_{NLos}$ are variables denoting the attenuation factors of the LoS and NLoS links, respectively. To ensure the communication success, the probability of an LoS link and an NLoS link connection between UAV $n$ and the ground user UE $i$ are respectively expressed as follows:

$$P_{Los} = \frac{1}{1 + aexp(-b(\frac{180}{\pi}\theta_{i,n} - a))} \tag{6}$$

$$P_{NLos} = 1 - P_{Los} \tag{7}$$

where $a$, $b$ are constant values associated with the environment. $\theta_{i,n}$ represents the elevation angle with $\theta_{i,n} = arctan(\frac{h}{\sqrt{(x_n-x_i)^2+(y_n-y_i)^2}})$. The UAV-to-UE average path loss and transmission rate can be determined as follows:

$$PL_{n,i} = P_{Los} \times PL_{n,i}^{Los} + P_{NLos} \times PL_{n,i}^{NLos}$$
$$= \frac{A}{1+aexp(-b(\frac{180}{\pi}\theta_{i,n}-a))} + 20log(\sqrt{(x_n - x_i)^2 + (y_n - y_i)^2 + h^2}) + B \tag{8}$$

$$C_{n,i} = \frac{B_0}{A_n}log_2(1 + SNR_{n,i}), j \neq i \tag{9}$$

$$A = \eta_{Los} - \eta_{NLos}, B = 20log(4\pi f/c) + \eta_{NLos} \tag{10}$$

Assume that the available bandwidth $B_0$ of UAVs is divided equally among users. The signal-to-noise ratio (SNR) is determined as follows:

$$SNR_{n,i} = \frac{P_n - PL_{n,i}}{\sigma}. \tag{11}$$

The total number of users associated with UAV $n$ is denoted by $A_n$. To account for the link quality between UAV $n$ and ground user UE $i$, a threshold $\sigma_{i,n}$ is set for constraint as Equation (10).

$$\sigma_{i,n} = \begin{cases} 1, & d_{i,n}^h \leq r_n \text{ and } SNR_{n,i} < SNR_{th} \\ 0, & d_{i,n}^h > r_n \text{ or } SNR_{n,i} \geq SNR_{th} \end{cases} \tag{12}$$

II.    UAV to UAV link

With regard to the link between UAVs, we rely on the research presented in [7], which establishes that the UAV-UAV channels are primarily influenced by the Line of Sight (LoS) component. While there may be some limited multi-path fading due to ground reflections, it has a negligible impact when compared to the effects experienced in UAV-ground or ground–ground channels. The communication between UAVs is mainly a line-of-sight link.

Consequently, the path loss of communication between UAVs can be attributed to the free space loss and expressed in the following manner:

$$PL_{n,m}^{Los} = 20log d_{n,m} + 20log f + 20log 4\pi/c \tag{13}$$

With respect to the issue of interference among UAVs [36], the assumption is that assigning different frequencies to different UAVs can help avoid annoyance and interference, which is not considered among UAVs in this paper. Hence, the signal-to-noise ratio and transmission rate between UAV $n$ and UAV $m$ are outlined below.

$$SNR_{n,m} = \frac{P_n - PL_{n,m}}{\sigma} \tag{14}$$

$$C_{n,m} = \frac{B_c}{|J_{1-n}|} log_2(1 + SNR_{n,m}) \tag{15}$$

where $P_n$ is the transmitting power of UAV $n$; $B_c$ is the communication bandwidth between UAV $n$ and UAV $m$, and $|J_{1-n}|$ is the number of one-hop nodes of UAVs.

III.  MBS to UAV link

When it comes to the connection between the UAV and the long-distance MBS, the signal strength varies according to the distance (which could span hundreds or thousands of wavelengths) and is directly proportional to the square of the distance due to the wave-energy diffusion phenomenon. This phenomenon is solely related to the transmission path, whereby the longer the path, the higher the path loss. Therefore, the path loss of communication between MBS and UAV $n$ is denoted as follows:

$$PL_{n,MBS}^{Los} = d_{n,MBS}^{-2}, \ PL_{n,MBS}^{NLos} = \xi d_{n,MBS}^{-2} \tag{16}$$

$$\theta_{n,MBS} = arctan(\frac{z_n}{\sqrt{(x_n - x_{MBS}) + (y_n - y_{MBS})^2}}) \tag{17}$$

$$P_{Los}^{n,MBS} = \frac{1}{1 + aexp(-b(\frac{180}{\pi}\theta_{n,MBS} - a))}, \ P_{NLos}^{n,MBS} = 1 - P_{Los}^{n,MBS} \tag{18}$$

$$PL_{n,MBS} = P_{Los}^{n,MBS} \times PL_{n,MBS}^{Los} + P_{NLos}^{n,MBS} \times PL_{n,MBS}^{NLos} \tag{19}$$

where $d_{n,MBS}$ denotes the distance between the MBS, and the coordinates of the MBS are $l_{MBS} = (x_{MBS}, y_{MBS})$. The signal-to-noise ratio and transmission rate between the MBS and UAV $n$ are illustrated below.

$$SNR_{n,MBS} = \frac{P_s - PL_{n,MBS}}{\sigma}, C_{n,MBS} = \frac{B_0}{N}log_2(1 + SNR_{n,MBS}) \tag{20}$$

where is $B_0$ is the back-haul network bandwidth, and $N$ is the number of drones accessing the MBS.

*3.4. Content Cache Model*

In a cache-enabled UAV-assisted network, each UAV is assumed to have a limited cache capacity and the ability to actively store a certain amount of content in its memory. All contents are accessible for reading in the MBS, and the file library is defined as $\mathcal{F} = \{1, 2 \cdots, F\}$. To simplify the analysis, the size of each file item is set to be equal, which is denoted as $M$ bits, and the UAV's storage is limited to $H$ bits. The file popularity is assumed to follow the Zipf distribution [37], with parameter $\kappa$ indicating the skewness of popularity.

$$P_f = \frac{i^{-\kappa}}{\sum_{l=1}^{F} l^{-\kappa}} \tag{21}$$

where $Q_f$ is the popularity of the content $f$. To increase the cache-hit probability, UAVs utilize a probability distribution of file popularity to cache specific files, and the file popularity of all items can be denoted as $[P_1, P_2, \cdots, P_f, \cdots P_F]$. The cache status of the UAV is described using a binary decision $\tau_{f,n}$. The value of $\tau_{f,n}$ equal to 1 represents that the requested file is stored in the UAV $n$, while the value equal to 0 indicates that the file is not cached in the UAV $n$.

### 3.5. MOS Model

To assess the users' satisfaction, we employ the Mean Opinion Score (MOS) model [38] to evaluate the quality of the network system service, which takes the transmission delay into account.

$$MOS_{i,n,f} = c_1 ln(1/D_{i,n,f}) + c_2 \tag{22}$$

where $D_{i,n,f}$ denotes the content transmission delay; the $c_1$ and $c_2$ are constants, and $c_1 = 1.3254$, $c_2 = 4.6746$. It is evident that a shorter transmission delay results in a higher $MOS_{i,n,f}$, which provides a better user experience. The various potential scenarios are elucidated below:

When the file $f$ is cached in UAV $n$, where $R(S_n)$ denotes the cache file set of UAV $n$, the transmission delay for UE $i$ to obtain the file is as shown below.

$$D_{i,n,f} = \frac{M}{C_{n,i}}, \ f \in R(S_n) \tag{23}$$

When the file is not cached in UAV $n$ or in its neighbor nodes, then it needs to get the demanded file from the MBS through the wireless link to satisfy the user's request, and the transmission delay for UE $i$ to obtain the file is expressed as below.

$$D_{i,n,f} = \frac{M}{C_{s,n}} + \frac{M}{C_{n,i}}, \ f \in R(S_{RBS}) \tag{24}$$

When the file is not cached in UAV $n$ but in its one-hop neighbor node $m$, and $m$ is the closest cache node to $n$ among all one-hop nodes, the transmission delay for UE $i$ to obtain the file is as follows:

$$D_{i,n,f} = \frac{M}{C_{n,i}} + \frac{M}{C_{n,m}}, f \in R(C_{J_{1-n}}) \tag{25}$$

If the file is not cached in UAV $n$ or its one-hop neighborhood node $m$, but in node $m$'s one-hop neighbor $m\prime$, and $m\prime$ is the node with the shortest path away from node $n$ among all two-hop cache nodes, the equation below showcases the transmission delay for UE $i$ to acquire the file.

$$D_{i,n,f} = \frac{M}{C_{i,n}} + \frac{M}{C_{n,m}} + \frac{M}{C_{m,m\prime}}, \ f \notin R(C_{J_{1-n}}), \ f \in R(C_{J_{2-n}}) \tag{26}$$

In conclusion, the transmission delay for UE $i$ to obtain the content from the associated UAV $n$ can be summarized as follows:

$$\begin{aligned}
D_{i,n,f} = \quad & \sigma_{i,n}[\tau_{f,n}\tfrac{M}{C_{i,n}} + (1-\tau_{f,n})\tau_{f,m}(\tfrac{M}{C_{i,n}} + \tfrac{M}{C_{n,m}}) + (1-\tau_{f,n})(1 \\
& -\tau_{f,m})\tau_{f,m\prime} \times (\tfrac{M}{C_{i,n}} + \tfrac{M}{C_{n,m}} + \tfrac{M}{C_{m,m\prime}}) + (1-\tau_{f,n})(1-\tau_{f,m})(1 \\
& -\tau_{f,m\prime})(\tfrac{M}{C_{s,n}} + \tfrac{M}{C_{i,n}})]
\end{aligned} \tag{27}$$

Considering the possibility of a delay that may be lower than 1, which results in a negative $MOS_{i,n,f}$, we revise the Equation (22). Therefore, the $MOS_{i,n,f}$ of UE $i$ appealing content $f$ from the associated UAV $n$ is expressed as follows:

$$MOS_{i,n,f} = P_{i,f} \times [c_1 ln(100/D_{i,n,f}) + c_2] \tag{28}$$

where $P_{i,f}$ is the probability that user $i$ requests for file $f$.

### 3.6. Problem Generation

In this section, we analyze the users' satisfaction and coverage of the whole system performance. Based on the given $MOS_{i,n,f}$, the $MOS$ for the whole system is expressed as follows:

$$MOS = \sum_{n \in N} \sum_{i \in I} \sum_{f \in F} MOS_{i,n,f} \tag{29}$$

Also, the user coverage of the system is expressed as the following:

$$p_{cover} = \sum_{i \in I} \sum_{n \in N} \sigma_{i,n} \tag{30}$$

Based on the above analysis, this paper takes both users' satisfaction and coverage into account. Consequently, the optimization problem under certain constraints can be defined as follows:

$$P: \ max(\alpha \ MOS + \beta p_{cover}) = max \left( \alpha \sum_{n \in N} \sum_{i \in I} \sum_{f \in F} \frac{MOS_{i,n,f}}{E_n} + \beta \sum_{i \in I} \sum_{n \in N} \sigma_{i,n} \right) \tag{31}$$

$$\text{s.t.} \begin{cases} \textbf{C1}: \|l_{m} - l_{n}\| \geq d_{min}, \forall m, n \in N, m \neq n \\ \textbf{C2}: \sigma_{i,n} = \{0,1\}, \forall n \in N, i \in I \\ \textbf{C3}: \sum_{n \in N} \sigma_{i,n} \leq 1 \\ \textbf{C4}: \tau_{f,n} = \{1,0\}, \forall n \in N, f \in F \\ \textbf{C5}: \sum_{f \in F} \sum_{n \in N} \tau_{f,n} \times S \leq H, \forall n \in N, f \in F \end{cases} \tag{32}$$

The defined optimization problem is constrained by the binary decision $\sigma_{i,n}$ and $\tau_{f,n}$. C3 means that each user only associates with a single UAV. C5 is set to limit the caching capacity of each UAV.

## 4. Location and Cache Strategy Based on Potential Game

Based on the optimization problem formulated in Section 2, we leverage potential game theory to present a novel approach for tackling the optimization problem outlined in Section 3.6, which aims to optimize both the cache file placement and the locations of UAVs to maximize the users' satisfaction and coverage. First, the UAV 3D position deployment is decoupled into 2D planar deployment and altitude optimization. To achieve this, we utilize the K-means algorithm to initialize the 2D placement, followed by the calculation of the optimal altitude that can meet the SNR threshold. Then, the potential game theory is applied to model the joint optimization of the users' satisfaction and coverage, in which the log-linear-learning algorithm is employed to obtain the optimal results.

### 4.1. UAV Placement of Altitude

To guarantee the connection probability between UE and UAV, the ground users are clustered into several subsets using the K-means [39] clustering algorithm. The process involves initializing *k* cluster centers, calculating the Euclidean distance between each user and the cluster center using Equation (33), and assigning each user to the closest cluster center. Therefore, the initial horizontal coordinates of UAVs are then determined as the center of each user cluster.

$$d_{n,i} = \sqrt{(x_n - x_i)^2 + (y_n - y_i)^2}, \ i \in I, \ n \in N \tag{33}$$

As discussed in references [6,40], the UAV coverage is related to the altitude of the UAV, and the height of the UAVs is uniformly set for facilitating the establishment of the two-hop network for cooperative caching. The received SNR of the user must exceed a

specific threshold value to ensure the quality of the user's service. From Equation (8), it is concluded that:

$$\frac{A}{1 + aexp(-b(\frac{180}{\pi}\theta_{i,n} - a))} + 20log(d_{n,i}) + B \le P_n - \sigma\gamma_{th} \tag{34}$$

In fact, the maximum path loss can be determined from the above equation as follows:

$$PL_{max} = \frac{A}{1 + aexp(-b(\frac{180}{\pi}\theta_{i,n} - a))} + 20log(d_{n,i}) + B = P_n - \sigma\gamma_{th} \tag{35}$$

To determine the optimal altitude point for achieving the best coverage, $h$ can be expressed in the following manner:

$$d_{n,i} = \frac{h}{sin\theta_{i,n}}, \ B = 20log(4\pi f/c) + \eta_{NLos} \tag{36}$$

$$h = 10exp[(P_n - \sigma\gamma_{th} - B - \frac{A}{1 + aexp(-b(\frac{180}{\pi}\theta_{i,n} - a))})\frac{1}{20}]sin\theta_{i,n} \tag{37}$$

Then, compute the first order partial derivatives of $h$ with respect to $\theta$, denoted as follows, respectively:

$$\frac{\partial h}{\partial \theta} = \frac{1}{20}ln10 \times 10^{(P_n - \sigma\gamma_{th} - B - \frac{A}{1 + aexp(-b(\frac{180}{\pi}\theta_{i,n} - a))})\frac{1}{20}} \times \frac{Aab\frac{180}{\pi}exp(-b(\frac{180}{\pi}\theta_{i,n} - a))}{[1 + aexp(-b(\frac{180}{\pi}\theta_{i,n} - a))]^2}$$
$$\times sin\theta_{i,n} + cos\theta_{i,n} \times 10^{(P_n - \sigma\gamma_{th} - \frac{A}{1 + aexp(-b(\frac{180}{\pi}\theta_{i,n} - a))})\frac{1}{20}} = 0 \tag{38}$$

In this case, the optimal altitude of the UAV is approximated with the determined values of UAV coverage radius and $\theta_{opt}$.

$$\frac{1}{20}ln10 \times \frac{Aab\frac{180}{\pi}exp(-b(\frac{180}{\pi}\theta_{i,n} - a))}{[1 + aexp(-b(\frac{180}{\pi}\theta_{i,n} - a))]^2}sin\theta_{i,n} + cos\theta_{i,n} = 0 \tag{39}$$

$$\frac{Aabexp(-b(\frac{180}{\pi}\theta_{i,n} - a))}{[1 + aexp(-b(\frac{180}{\pi}\theta_{i,n} - a))]^2} + \frac{\pi}{9ln10}tan\theta_{i,n} = 0 \tag{40}$$

As $A, a$ and $b$ are known parameters, we could calculate the optimal angle $\theta_{opt}$ from Equation (37). Then, the optimal altitude is derived from Equation (34).

### 4.2. Joint Strategy for UAV Cooperative Caching and 2D Deployment
#### 4.2.1. Potential Game

Formally, the game is denoted as $\mathcal{G} = \{\mathcal{N}, \{\mathcal{T}_n\}_{\mathcal{N}}, \{\mathcal{A}_n\}_{\mathcal{N}}, \{u_n\}_{\mathcal{N}}\}$, in which UAVs act as the players. $\mathcal{N} = \{1, 2, 3, \cdots N\}$ is the set of UAVs. The set $\mathcal{T}_n$ consists of all two-hop neighbors of UAV $n$. $\mathcal{A}_n$ is a set of the available actions for UAV $n$, and $u_n$ is the utility of UAV $n$. The system utility function is denoted as $U_n(a_n, a_{-n})$, where $a_n = \{f_1, \cdots, f_K, l_n\}$ is the action of UAV $n$ incorporating the file-caching strategy and 2D location, and $a_{-n}$ represents the action profile of all UAVs except UAV $n$ with respect to file-caching strategy and 2D location. The Nash equilibrium and the exact potential game of the formulated game are defined as follows.

**Definition 1.** (Nash equilibrium): *For a strategic game $\mathcal{G}$, a joint cache and horizontal action profile $a^* = \{a_1^*, \ldots, a_n^*\}$ is a mixed strategy NE if and only if no UAV can improve its utility by unilaterally deviating its strategy while the others keep theirs unchanged. Then the action profile $a^*$*

*is said to be a NE of the game $\mathcal{G}$. If the inequality is strict, then the action profile $a^*$ is said to be a strict NE of the game $\mathcal{G}$.*

$$U_n(a_n^*, a_{-n}^*) \geq U_n(a_n, a_{-n}^*), \forall n \in N, \ a_n \in \mathcal{A}_n, a_n \neq a_n^* \tag{41}$$

**Definition 2.** (Exact potential game): *If there exists a potential function satisfying with the following equation $\Phi : \mathcal{A}_1 \otimes \mathcal{A}_2 \otimes \ldots \otimes \mathcal{A}_n \mapsto R$, when UAV $n$ changes its strategy ($a_n \longrightarrow a_{n'}$), the increment of its game utility $\Phi$ is equal to that of the potential function $U_n$, then it is an exact potential game (EPG). The exact potential game model has at least one pure Nash equilibrium solution.*

$$\Phi(a_n^*, a_{-n}) - \Phi(a_n, a_{-n}) = U_n(a_n^*, a_{-n}) - U_n(a_n, a_{-n}) , \ \forall n \in N,$$
$$a_n, \ a_n^* \in \mathcal{A}_n \tag{42}$$

Equation (37) indicates that the change in the utility function due to the unilateral deviation of a single UAV is identical to the change in the potential function. According to the utility theory, the purpose of the utility function is to assess the individual performance of each UAV while maintaining a reasonable trade-off between the MOS and coverage. It can be achieved by rationalizing the cache placement and location. The defined utility function is expressed as follows.

$$
\begin{aligned}
U_n(a_n, a_{\mathcal{T}_n}) &= u_n(a_n, a_{\mathcal{T}_n}) + \sum_{k \in \mathcal{T}_n} u_k(a_k, a_{\mathcal{T}_k}) - (u_n(a_0, a_{-n}) \\
&\quad + \sum_{k \in \mathcal{T}_n} u_k(a_k, a_{\mathcal{T}_k \setminus n})) \\
&= \alpha \sum_{i \in I} \sum_{f \in F} MOS_{i,n,f} + \beta \sum_{i \in I} \sigma_{i,n} + (\alpha \sum_{k \in \mathcal{T}_n} \sum_{f \in F} MOS_{i,n,f} \\
&\quad + \beta \sum_{i \in I} \sigma_{i,n}) - (u_n(a_0, a_{-n}) + \sum_{k \in \mathcal{T}_n} u_k(a_k, a_{\mathcal{T}_k \setminus n}))
\end{aligned}
\tag{43}
$$

where $a_{\mathcal{T}_n}$ is the strategy profile of UAV $n$'s two-hop neighbors, and $a_0$ means that UAV $n$ does not provide any service for any UE. Thus $a_{\mathcal{T}_k \setminus n}$ is the strategy profile of UAV $k$'s two-hop neighbors when UAV $n$ gives up working.

To optimize the problem P, the total potential function is defined as follows:

$$
\begin{aligned}
\Phi(a_1, &\ldots, a_{n-1}, a_n, \ a_{n+1}, \ldots a_N) \\
&= \sum_{n \in N} u_n(a_n, a_{\mathcal{T}_n}) \\
&= \sum_{n \in N} (\alpha \sum_{i \in I} \sum_{f \in F} MOS_{i,n,f} + \beta \sum_{i \in I} \sigma_{i,n})
\end{aligned}
\tag{44}
$$

**Theorem 1.** *The proposed joint strategy for UAV cooperative caching and the 2D deployment model is an exact potential game for which there exists at least one stable Nash equilibrium solution.*

**Proof of Theorem 1.** Based on Equation (44), the alteration attributed to the unilateral change in the individual utility of UAV $n$ can be expressed as follows.

$$
\begin{aligned}
\Phi(a_{n'}, a_{-n}) \quad &- \Phi(a_n, a_{-n}) \\
&= \sum_{n \in \mathcal{T}_n' \cup \{n'\} \cup N \setminus \mathcal{T}_n' \setminus \{n'\}} u_n(a_{n'}, \mathcal{T}_n') \\
&- \sum_{n \in \mathcal{T}_n \cup \{n\} \cup N \setminus \mathcal{T}_n \setminus \{n\}} u_n(a_n, \mathcal{T}_n)
\end{aligned}
\tag{45}
$$

where $a_{n'}$ is derived from $a_n$ by changing UAV $n$'s cache placement or 2D position strategy. The change of the potential function caused by the unilateral change is demonstrated below.

If it is assumed that the UAV $n$ undergoes one-step changes at a time, only one alternative for caching placement or 2D position can be modified at a time. As a result, there are possible scenarios that result in action changes.

**Case 1**: The first scenario is when a cached content in UAV $n$ is changed while the 2D position remains unchanged. The detailed proof of the potential function change is demonstrated below.

$$
\begin{aligned}
\Phi(a_{n'}, a_{-n}) \quad &-\Phi(a_n, a_{-n}) \\
&= u_n(a_{n'}, \mathcal{T}_{n'}) - u_n(a_n, \mathcal{T}_n) \\
&+ \sum_{k' \epsilon \mathcal{T}_n'} u_{k'}(a_{k'}, \mathcal{T}_{k'}) - \sum_{k \epsilon \mathcal{T}_n} u_k(a_k, \mathcal{T}_k) \\
&+ \sum_{k' \epsilon N \backslash \mathcal{T}_n', \, k' \neq n} u_{k'}(a_{k'}, \mathcal{T}_{k'}) - \sum_{k \epsilon N \backslash \mathcal{T}_n, \, k \neq n} u_k(a_k, \mathcal{T}_k)
\end{aligned}
\tag{46}
$$

where $a_{-n} = (a_1, \ldots, a_{n-1}, \ldots, a_{n+1}, \ldots a_N)$, and $u_n(a_{n'}, \mathcal{T}_{n'})$ denotes the changed utility of UAV $n$ after unilaterally taking the action $a_{n'}$.

Considering that the UAV $n$'s alteration of the caching file impacts solely the utility of its two-hop neighbors, the following equation can be concluded.

$$
\sum_{k' \epsilon N \backslash \mathcal{T}_n', \, k' \neq n} u_{k'}(a_{k'}, \mathcal{T}_{k'}) - \sum_{k \epsilon N \backslash \mathcal{T}_n, \, k \neq n} u_k(a_k, \mathcal{T}_k) = 0
\tag{47}
$$

Therefore, based on Equations (44), (46) and (47), it can be inferred that if the UAV $n$ takes a unilateral action, the alteration in the utility function presented in Equation (43) is equivalent to the modification in the potential function detailed in Equation (45).

$$
\begin{aligned}
\Phi(a_{n'}, a_{-n}) \quad &-\Phi(a_n, a_{-n}) \\
&= [u_n(a_{n'}, \mathcal{T}_{n'}) + \sum_{k' \epsilon \mathcal{T}_n'} u_{k'}(a_{k'}, \mathcal{T}_{k'}) - (u_n(a_0, a_{-n}) \\
&+ \sum_{k \in \mathcal{T}_n} u_k(a_k, a_{\mathcal{T}_k \backslash n}))] - \{u_n(a_n, \mathcal{T}_n) + \sum_{k \epsilon \mathcal{T}_n} u_k(a_k, \mathcal{T}_k) \\
&- (u_n(a_0, a_{-n}) + \sum_{k \in \mathcal{T}_n} u_k(a_k, a_{\mathcal{T}_k \backslash n}))\} \\
&= U_n(a_{n'}, a_{\mathcal{T}_{n'}}) - U_n(a_n, a_{\mathcal{T}_n})
\end{aligned}
\tag{48}
$$

**Case 2**: When the 2D position of UAV $n$ is changed from $l_n$ and changes to $l_{n'}$, the set of two-hop neighbors of the UAV $n$ becomes $\mathcal{T}_{n'}$, and the detailed proof of the change about the potential function is demonstrated as follows:

$$
\begin{aligned}
\Phi(a_{n'}, a_{-n}) - \Phi(a_n, a_{-n}) \quad &= \sum_{n' \in Z \cup \mathcal{T}_{n'} \backslash Z \cup Y \cup N \backslash \mathcal{T}_{n'} \backslash \{n'\} \backslash Y \cup \{n'\}} u_n(a_{n'}, \mathcal{T}_{n'}) \\
&- \sum_{n \in Z \cup \mathcal{T}_n \backslash Z \cup Y \cup N \backslash \mathcal{T}_n \backslash \{n\} \backslash Y \cup \{n\}} u_n(a_n, \mathcal{T}_n) \\
&= u_n(a_{n'}, \mathcal{T}_{n'}) + \sum_{n' \in Z \cup \mathcal{T}_{n'} \backslash Z \cup Y \cup N \backslash \mathcal{T}_{n'} \backslash \{n'\} \backslash Y} u_n(a_{n'}, \mathcal{T}_{n'}) \\
&- [u_n(a_n, \mathcal{T}_n) + \sum_{n \in Z \cup \mathcal{T}_n \backslash Z \cup Y \cup N \backslash \mathcal{T}_n \backslash \{n\} \backslash Y} u_n(a_n, \mathcal{T}_n)]
\end{aligned}
\tag{49}
$$

Let $Z = \mathcal{T}_{n'} \cap \mathcal{T}_n, Y = N \backslash \mathcal{T}_{n'} \backslash \{n'\} \cap N \backslash \mathcal{T}_n \backslash \{n\}$, then it could be found that $N = Z \cup \mathcal{T}_{n'} \backslash Z \cup Y \cup N \backslash \mathcal{T}_{n'} \backslash \{n'\} \backslash Y \cup \{n'\}$. Likewise, it could be obtained that $N = Z \cup \mathcal{T}_n \backslash Z \cup Y \cup N \backslash \mathcal{T}_n \backslash \{n\} \backslash Y \cup \{n\}$. Based on the above analysis, the subsequent equation has been established.

$$
\sum_{n' \in Y} u_n(a_{n'}, \mathcal{T}_{n'}) - \sum_{n \in Y} u_n(a_n, \mathcal{T}_n) = 0
\tag{50}
$$

where $N\backslash\mathcal{T}_{n'}\backslash\{n'\}\backslash Y = \mathcal{T}_n\backslash Z$, $N\backslash\mathcal{T}_n\backslash\{n\}\backslash Y = \mathcal{T}_{n'}\backslash Z$. Since UAV $n$'s action only affects the utility of its specific two-hop neighbors, UAV$n'$ in the set $\mathcal{T}_n\backslash Z$ and $n$ in the set $\mathcal{T}_{n'}\backslash Z$ are not benefited by UAV $n$'s individual changes. As a result, we can conclude that:

$$
\begin{aligned}
\Phi(a_{n'}, a_{-n}) \quad &- \Phi(a_n, a_{-n}) \\
&= u_n(a_{n'}, \mathcal{T}_{n'}) + \sum_{n' \in Z \cup \mathcal{T}_{n'} \backslash Z \cup \mathcal{T}_n \backslash Z} u_n(a_{n'}, \mathcal{T}_{n'}) - [u_n(a_n, \mathcal{T}_n) \\
&+ \sum_{n \in Z \cup \mathcal{T}_n \backslash Z \cup \mathcal{T}_{n'} \backslash Z} u_n(a_n, \mathcal{T}_n)] \\
&= [u_n(a_{n'}, \mathcal{T}_{n'}) \\
&+ \sum_{k' \epsilon \mathcal{T}_{n'}} u_{k'}(a_{k'}, \mathcal{T}_{k'}) + \sum_{n' \in \mathcal{T}_n \backslash Z} u_n(a_{n'}, \mathcal{T}_{n})_{'}] - [u_n(a_n, \mathcal{T}_n) \\
&+ \sum_{n \in \mathcal{T}_n} u_n(a_n, \mathcal{T}_n) + \sum_{n \in \mathcal{T}_{n'} \backslash Z} u_n(a_n, \mathcal{T}_n) \ ] \\
&= U_n(a_{n'}, a_{\mathcal{T}_{n'}}) - U_n(a_n, a_{\mathcal{T}_n})
\end{aligned}
\tag{51}
$$

From Equations (46)–(51), we can find that when UAV $n$ takes a unilateral action; the change in utility function given in (43) is the same as the change in the potential function in (44).

$$
\Phi(a_{n'}, a_{-n}) - \Phi(a_n, a_{-n}) = U_n(a_{n'}, a_{\mathcal{T}_{n'}}) - U_n(a_n, a_{\mathcal{T}_n})
\tag{52}
$$

According to Definition 2, the proposed game model is an exact potential game with at least one Nash equilibrium. Moreover, the solution to the maximization problem **P** also serves as a Nash equilibrium. Therefore, the game model satisfies the conditions of an exact potential game. $\square$

4.2.2. Log-Linear-Learning Algorithm

To address the problem in the game, it is essential to develop an uncoupled learning approach that has minimal reliance on interaction information, ensures rapid convergence, and can be efficiently distributed to achieve the globally optimal solution. The log-linear-based learning rule, a well-established scheme known for attaining optimal policy solutions, has been successfully employed to address this problem, as referenced in [41–44] (Algorithm 1).

---

**Algorithm 1. Joint Strategy for UAV Cooperative Caching and 2D Deployment**

---

**Input**: The set of UEs $\{1, 2, 3, \cdots I\}$,
**Output**: UAV horizontal location $\{l_n\}_\mathcal{N}$, content cache strategy.
**Initialization**: Initialize $a_n$ ($\forall n \in N$), while the initial 2D position of UAVs are the center of the clusters, respectively; set the number of iterations $t = 1$, $t \leq t_{max}$, $t_{max}$ as the maximum round.
**Step 1**: All UAVs exchange information of current action $a_n$ with their two-hop neighbor nodes UAVs.
**Step 2**: Randomly select UAV $n$. Then, for the selected UAV $n$, calculate its utility function $U_n^{(t)}(a_n, a_{-n})$ by (37).
**Step 3**: The selected UAV $n$ randomly chooses an action $a_{n'} \in \mathcal{A}_n$ with equal probability and keeps all the other UAVs' action profile unchanged. Then it calculates the utility function based on the selection $a_{n'}$, denoted as $U_{n'}(a_{n'}, a_{-n})$.
**Step 4**: The selected UAV $n$ adheres to the following rules to update its selection in iteration $t + 1$, which means the probability of choosing action $a_{n'}$ in iteration $t + 1$ is $P_{a_m}(t + 1)$.

$$
P_{a_{n'}}(t+1) = \frac{exp\{vU_n(a_{n'}, a_{-n}(t))\}}{\sum_{\overline{a_{n'}} \in \mathcal{A}_n} exp\{vU_n(\overline{a_{n'}}, a_{-n}(t))\}}
\tag{53}
$$

Where $v$ is the learning parameter.
**Step 5**: If $P_{a_m}(t + 1) = 0.99$ or $t = t_{max}$, then stop the iteration; otherwise, let $t = t + 1$, go to Step 2.

---

*4.3. Algorithm Convergence Analysis*

In the scheme designed above, at every time step t, one UAV $n \in \mathcal{N}$ is randomly selected. It updates its state by choosing one action $a_{n'}$ from action set $\mathcal{A}_n$ according to the Boltzmann distribution with parameter $v \geq 0$. The action $a_{n'}$ is selected with probability $P_{a_{n'}}(t)$, which is denoted as follows.

$$p_{a_{n'}}(t) = \frac{exp\{vU_n(a_{n'}, a_{-n}(t-1))\}}{\sum_{\overline{a_{n'}} \in \mathcal{A}_n} exp\{vU_n(\overline{a_{n'}}, a_{-n}(t-1))\}} \tag{54}$$

The above statement clearly establishes a Markov chain with a state space equivalent to the set $\mathcal{A}$ of all joint actions.

$$\mathcal{A} = \mathcal{A}_1 \otimes \mathcal{A}_2 \otimes \ldots \otimes \mathcal{A}_n \tag{55}$$

Moreover, the logit dynamics can be defined as follows.

**Definition 3.** *Let space $\mathcal{A} = \prod_{n \in N} \mathcal{A}_n$ be the state space of all actions of the UAV sets $\mathcal{N}$, and the combination of all node actions is $a(t) = \{a_n(t)\}$ $(n \in N)$, and the state transfer payoff is $U_n(a_{n'}, a_{-n}(t))$. $M_v = \{\mathcal{A}, a(t), P(t)\}$ on the joint actions $\mathcal{A}$ ith transition probability P is the homogeneous Markov chain.*

$$P(a_1, a_2) = \frac{1}{N} \begin{cases} P_{a_n^2}(t) \,, \ a_{-n}^2 = a_{-n}^1 \ and \ a_1 \neq a_2 \\ \sum_{n \in N} P_{a_n^2}(t), \ a_1 = a_2 \end{cases} \tag{56}$$

*where $a_1$ and $a_2$ are the assumed state, and $a_1, a_2 \in A$. Where $a_n^1$ represents the action profile of UAV n that facilitates the transition from the current state into state 1, $a_n^2$ is the action profile of UAV n that facilitates the transition from the current state into state 2.*

The proof for the unique distribution in the potential game is as follows:

**Theorem 2.** *If $M_v = \{\mathcal{A}, a(t), P(t)\}$ is a potential game with utility function $\Phi$, then the Markov chain is a unique stationary distribution $\pi$ with*

$$\pi(a) = \frac{exp\{v\Phi(a)\}}{\sum_{S \in A} exp\{v\Phi(S)\}} \tag{57}$$

**Proof of Theorem 2.** According to the detailed balance condition for Markov chains: given a Markov chain, a distribution $\pi$ and a probability transfer matrix **P**, if the following formula is satisfied, then this Markov chain has a stationary distribution.

$$\pi(a_1)P_{a_1 \, a_2} = \pi(a_2)P_{a_2 \, a_1} \tag{58}$$

It is clearly fulfilled in Equation (58) if $a_1$ equal to $a_2$, and if $a_1$ and $a_2$ differ in more by the $n$th UAV, the detailed proof is demonstrated as follows.

Assuming that the state distribution at the current time step is $\pi(a_1)$, the state is transferred to $\pi(a_2)$ with the probability of $P_{a_1 a_2}$. As the proposed algorithm selects only one UAV randomly to update its action, only the certain UAV's state is transferred between $a_1$ and $a_2$.

Let $a'_n$ denote the strategy of UAV $n$, $a'_n \in \mathcal{A}_n$, then the state distribution of the UAV's changing from $\pi(a_1)$ with transfer probability $P_{a_1 a_2}$ is denoted as follows:

$$
\begin{aligned}
\pi(\,a_1)P_{a_1\,a_2} &= \frac{1}{N}\frac{exp\{v\Phi(\,a_1)\}}{\sum_{S\in A}exp\{v\Phi(S)\}}P_{a_n^2}(t) \\
&= \frac{1}{N}\frac{exp\{v\Phi(\,a_1)\}}{\sum_{S\in A}exp\{v\Phi(S)\}}\frac{exp\{vU_n(a_n^2,a_{-n}^2(t-1))\}}{\sum_{\overline{a_{n'}}\in\mathcal{A}_n}exp\{vU_n(\overline{a_{n'}},a_{-n}(t-1))\}} \\
&= \frac{1/N(exp\{v\Phi(\,a_1)+vU_n(a_n^2,a_{-n}^2(t-1)\})}{\sum_{S\in A}exp\{v\Phi(S)\}\sum_{\overline{a_{n'}}\in\mathcal{A}_n}exp\{vU_n(\overline{a_{n'}},a_{-n}(t-1))\}}
\end{aligned}
\tag{59}
$$

Similarly, it could be obtained as follows:

$$
\pi(\,a_2)P_{a_2\,a_1} = \frac{1/N(exp\{v\Phi(\,a_2)+vU_n(a_n^1,a_{-n}^1(t-1)\})}{\sum_{S\in A}exp\{v\Phi(S)\}\sum_{\overline{a_{n'}}\in\mathcal{A}_n}exp\{vU_n(\overline{a_{n'}},a_{-n}(t-1))\}}
\tag{60}
$$

Considering the potential game theory presented in Equation (42), it can be inferred that:

$$
v(\Phi(\boldsymbol{a_1})-\Phi(\boldsymbol{a_2})) = v(U_n(a_n^1,a_{-n}^1)-U_n(a_n^2,a_{-n}^2))
\tag{61}
$$

Therefore, it can be presented to demonstrate the equation below, which is exactly the balanced stationary equation of the Markov process [43]. Then, its stationary distribution is characterized by Equation (57) since the above algorithm has a unique stationary distribution, and the distribution satisfies the balanced equations of its Markov process.

$$
\pi(\,a_1)P_{a_1\,a_2} = \pi(\,a_2)P_{a_2\,a_1}
\tag{62}
$$

Hence, Theorem 2 is demonstrate; thus the above log-linear learning algorithm can be converged and established to possess asymptotic optimality. □

## 5. Performance Evaluation

In this section, we perform a series of numerical simulations and evaluate the proposed Joint Strategy for UAV Cooperative Caching placement and 2D Deployment scheme for the UAV deployment.

### 5.1. Parameter Setting

In the simulation, the network parameters are summarized in Table 1. The coverage area of the network is assumed to be a square region measuring 3000 m × 3000 m, with ground users randomly distributed throughout the area. Inspired by reference [6,30], the parameters are set as presented in Table 3. To evaluate the effectiveness of the proposed two-hop-based algorithm, we compare it with benchmark algorithms. These benchmark algorithms are defined as follows:

1.  one-hop-based algorithm: the UAV provides the service for ground users with sharing cooperation of its one-hop neighbors. In fact, we have achieved the implementation of the one-hop-based algorithm using the framework of K-means, along with the one-hop mechanism and log-linear-learning method.
2.  Non-cooperative cache-based algorithm: the UAV provides the service with its own cache or with the help of the MBS. This is the implementation of the non-cooperative cache-based algorithm using the framework of K-means, along with the log-linear-learning method.

We assume file caching randomly in the three schemes above according to the same content popularity distribution.

**Table 3.** Simulation Parameters.

| Parameter | Value | Parameter | Value |
|---|---|---|---|
| Target region | 3 km × 3 km | $\eta_{Los}$ | 1.6 dBm |
| Altitude h | 500 m | $\eta_{NLos}$ | 23 dBm |
| Total bandwidth $B_0$ | 20 MHZ | Environmental parameter a | 9.6177 |
| Learning parameter $\upsilon$ | 0.01 | Environmental parameter b | 0.28 |
| File size M | 10 M | learning parameter | 0.1 |
| UAV cache capability H | 30 M | UAV minimum safe distance | 100 m |
| Carrier frequency $f$ | 5 GHZ | UAV communication distance | 800 m |
| The MBS's position $(x_{MBS}, y_{MBS})$ | (10,000,10,000) | $\alpha$ | 1 |
| UAV transmit power | 20 dBm | $\beta$ | 100 |
| MBS transmit power | 43 dBm | Zipf parameter $\kappa$ | 0.6 |
| communication bandwidth $B_c$ | 20 MHZ | Gaussian Noise $\sigma$ | −80 dBm |
| Excessive pathloss coefficient $\xi$ | 100 | Threshold of communication $\gamma_{th}$ | 0.1 |

*5.2. Performance of the Proposed Scheme*

The relationship between the utility and varying the number of users is described in Figure 2, where the number of UAVs is 15, and there are 40 item files. As shown in Figure 2, the performance of our proposed two-hop cooperative-caching strategy is compared with that of one-hop-based caching strategy and a strategy without cooperative caching (only consider the random-cached file in the associated file). It can be seen that the total system utility of the one-hop and the proposed scheme is better than that of the random-base algorithm. The reason is that the former two strategies reduce the additional latency associated with requesting content from the base station over the back-haul network. Moreover, the performance of the proposed scheme is better than that of the one-hop strategy. The two-hop strategy has been developed to effectively share cached content with the two-hop neighborhood, enhancing the efficiency of content caching in UAVs for users. Moreover, it aims to strike a balance between transmission delay and coverage. It is observed that as the number of ground users increases, the system utility of all three methods also increases. This phenomenon can be attributed to the fact that the utility is the combined measure of users' MOS and coverage. The increase in users ultimately results in the expansion of the service number of associated UAVs in the cluster, consequently leading to a higher number of users being served by each given UAV. In addition, the probability of the shared cache being utilized also increases due to the greater number of users.

The relationship between the utility of the whole system and varying the number of UAVs is described in Figure 3, where the number of users is 150, and there are 20 item files. As shown in the figure, the proposed scheme performs better than those of other algorithms tested. As the number of UAV increases, system performance will increase accordingly because the more users are covered and served as the number of drones increases, and the two-hop shared cache files can be utilized effectively by more users.

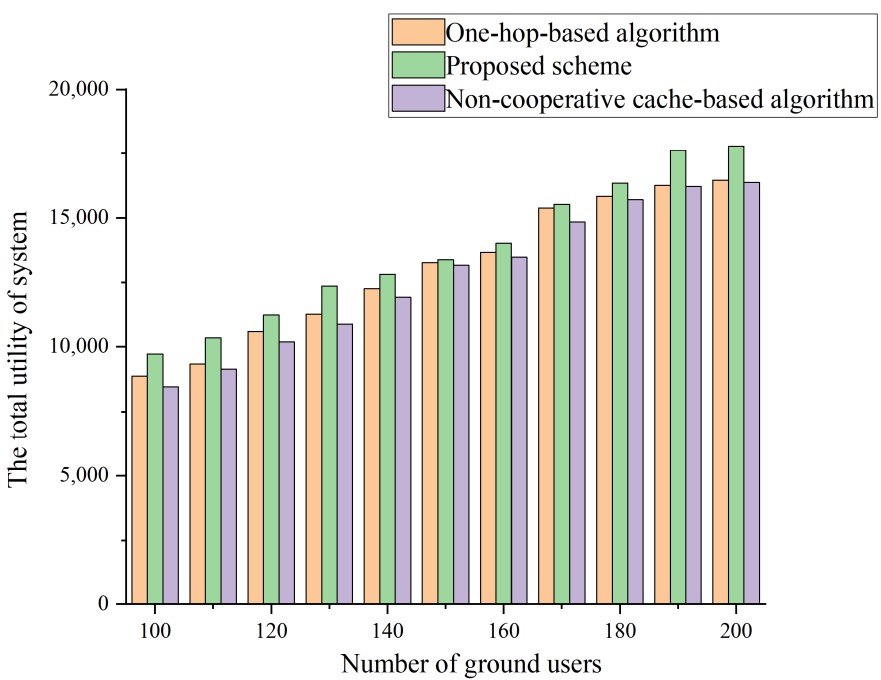

**Figure 2.** The total utility of the system with varying numbers of ground users.

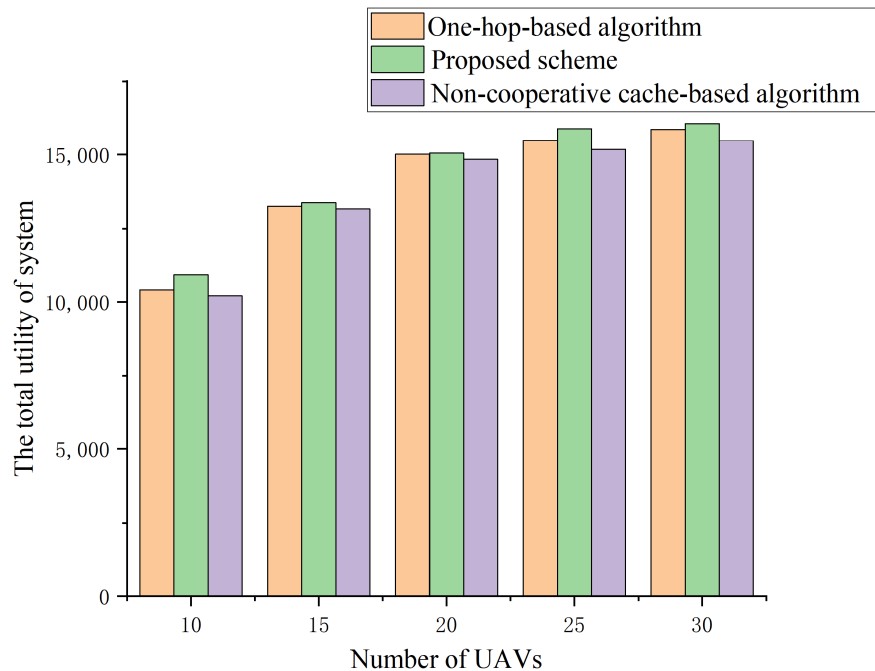

**Figure 3.** The total utility of system with varying numbers of UAVs.

Coverage ratio and MOS for various numbers of UAVs are represented in Figures 4 and 5, respectively, where the number of users is 150, and there are 20 item files. It can be seen in Figure 4 that when the number of UAVs increases, the coverage ratio increases. However, the coverage ratio of the one-hop-based algorithm and the random-cache algorithm are gradually approaching the proposed scheme while still lower than that of the proposed. We can find that when the number of UAVs increases, the overall trend of the one-hop-based algorithm and the proposed scheme are obviously better than non-cooperative cache-based algorithm. However, the MOS of the system of the one-hop-based algorithm is larger than that of the proposed one when the UAV number is 25 in Figure 5. The essence of the proposed algorithm is to utilize the advantage of gathering UAV groups, which naturally

leads to the decrease of coverage. Therefore, the utility function is designed to make a balance between the coverage and *MOS*. That is why sometimes the *MOS* of the system of the proposed scheme is lower than the one-hop-based algorithm.

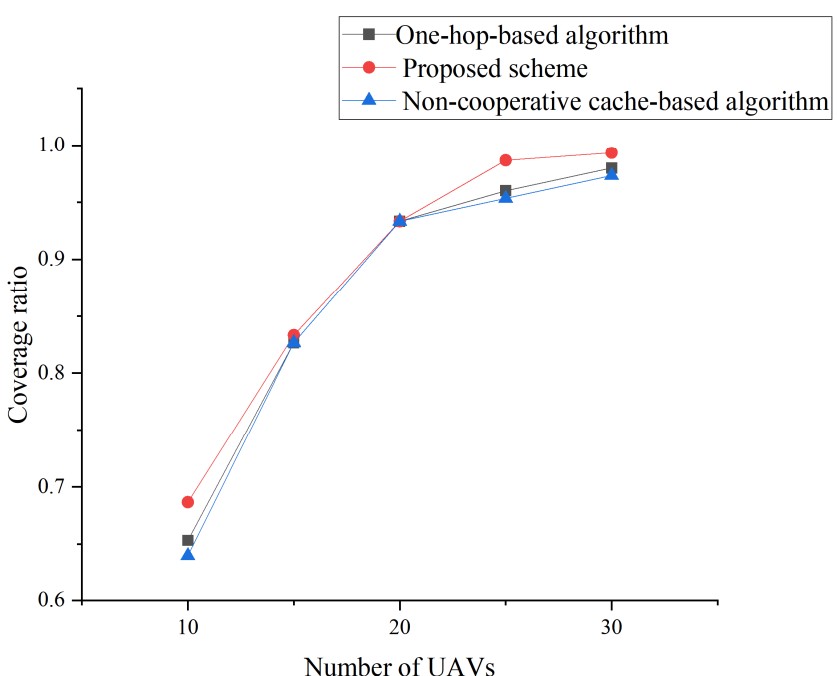

**Figure 4.** Impact of increasing UAVs on coverage ratio.

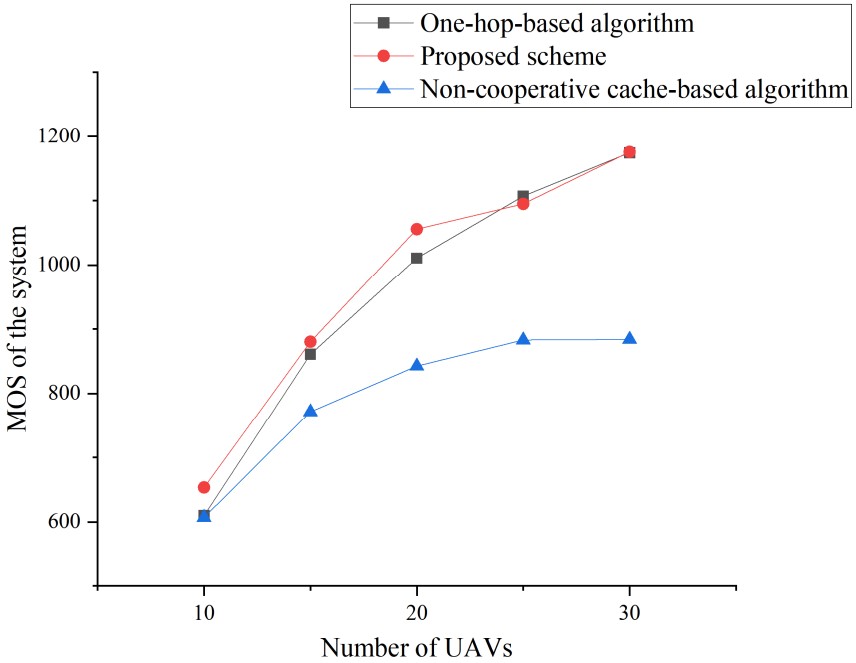

**Figure 5.** Impact of increasing UAVs on coverage ratio.

The total utility of the system for various file items is represented in Figure 6, where the cache capacity of each UAV is 3 items, the number of UAVs is set to be 20, and the number of users is 150. The simulation results show that the proposed algorithm achieves the highest system utility out of all three algorithms because the proposed scheme can fully utilize the neighborhood UAVs' caches. As the file items increase, the overall trend of the three schemes is down accordingly because the probability of the cache hit decreases.

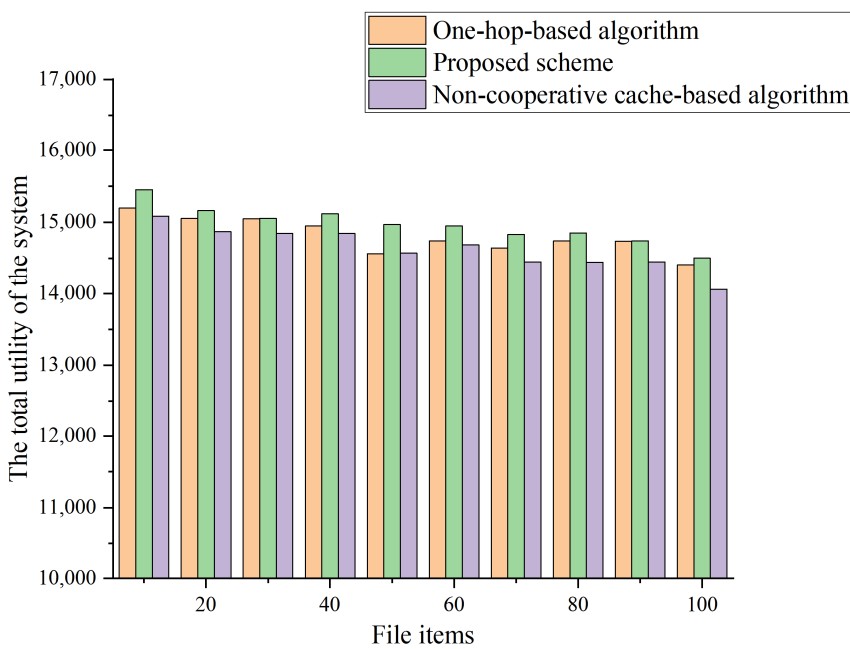

**Figure 6.** The total utility of the system with varying numbers of file items.

Figures 7 and 8 display the coverage and MOS values for varying numbers of file items. Upon examination of Figure 7, it is apparent that an increase in cache file items results in the sacrifice of coverage to maintain communication delays, leading to a slight decrease in the coverage rate. Additionally, the proposed scheme exhibits a flatter coverage trend compared to those of the one-hop-based algorithm and non-cooperative cache-based scheme, both of which do not perform as well. This is primarily because the coverage effect is heavily influenced by the number of UAVs, users, and their distribution.

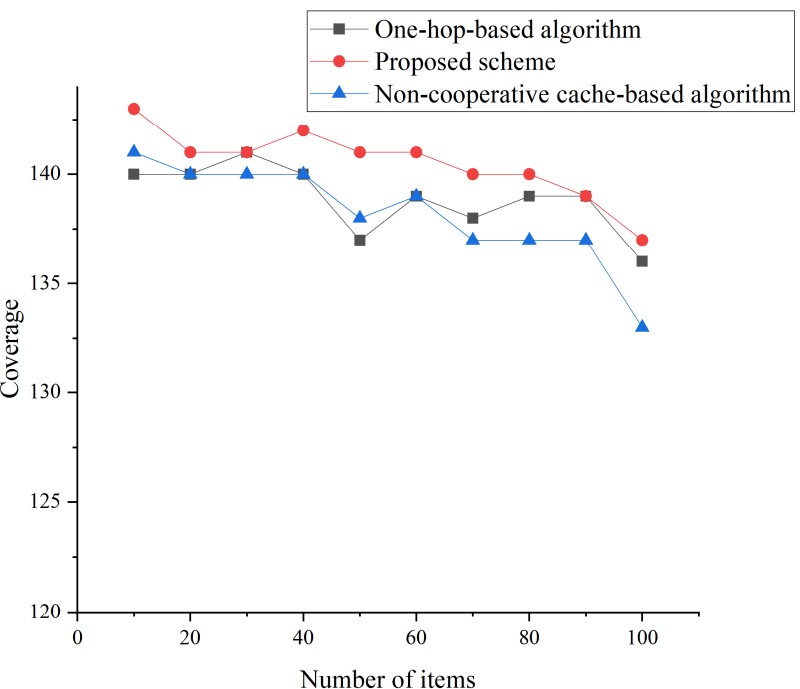

**Figure 7.** Impact of increasing file items on coverage.

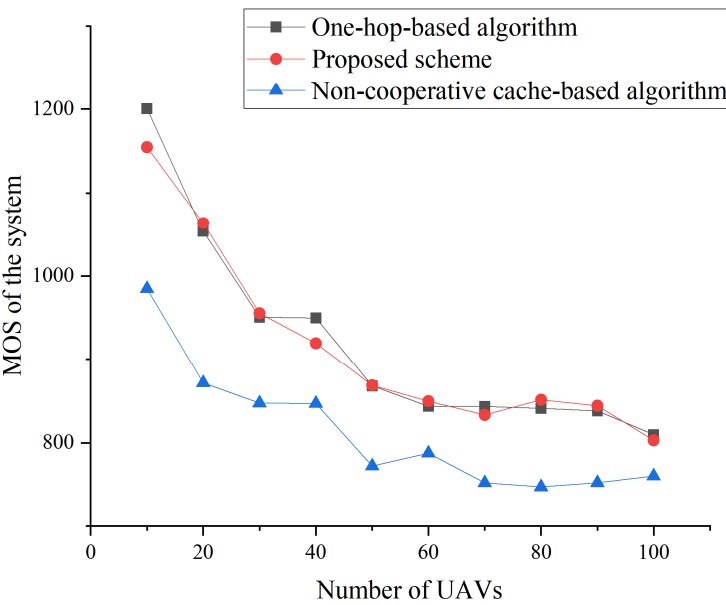

**Figure 8.** Impact of increasing UAVs on MOS.

　　　Considering the probability of the cache items of the UAVs and user requirements, the effect fluctuates when the number of file items changes in Figure 8. However, the proposed scheme has a better permanence on MOS than the other schemes tested when the file items increase.

*5.3. Convergence and Complexity Analysis*

　　　Currently, numerous scholars are utilizing the game theory model for cache placement and deployment in UAV-assisted networks. Consequently, this paper's approach is scrutinized against other conventional cooperative caching mechanisms in the existing literature.

　　　Table 4 demonstrates that the literature [22] just gave a brief procedure about a mixed-strategy equilibrium solution while the proposed method proves the existence and convergence of the equilibrium. Moreover, refs. [22,30] fail to consider the principle of the locality coupling effect, whereas the design of the utility function in this paper uniquely addresses this issue. Our proposed method meticulously describes the impact of cache placement or deployment on the utility of neighboring nodes, which can be adopted in various sequences.

**Table 4.** Methods comparison about game theory.

| Method | Scene Description | Method Versatility | Equilibrium Solution | Coupling Metrics | Action Space | Convergence | Complexity |
|---|---|---|---|---|---|---|---|
| Reference [22] | Detailed | General | Brief | Not considered | Space coordinates and cache files | Converged | $O(N)(O(C1) + O(S*C1*N) + O(C2))$ |
| Reference [30] | Detailed | Good | Detailed | Not considered | Choice of coalition | Converged | $O(N)(O(D1) + O(S*D1*N) + O(D2) + O(D3))$ |
| Proposed Approach | Detailed | Good | Detailed | Considered | Displacement in four directions and cache change | Converged | $O(N)(O(E1) + O(S*E1*K) + O(E2) + O(E3))$ |

　　　In reference [22], step 1 involves a complexity of $O(N)$ for selecting any UAV, where $N$ is determined by the number of UAVs. Furthermore, once a UAV senses the utility under this partition, the complexity is expressed as $O(C1)$, where $C1$ is a small constant determined by the calculation method specified in Equation (63). For steps 2 and 3, since the utility calculation of each action taken by the UAV $n$ needs to calculate the changed

utility value of all UAVs as depicted in Equation (63), calculating the changed utility value for all actions taken by the selected UAV $n$ has a complexity of $O(S * C1 * N)$, where $S$ is the strategy space. In step 4, after the utility values are compared, the complexity for UAV to update its action is $O(C2)$, where $C2$ is a small constant. Finally, the complexity of reference [22] can be represented by Equation (64).

$$U_n(a_n, a_{J_n}) = R(a_n, a_{-n}) - R(a_{n=0}, a_{-n}) \tag{63}$$

$$C_{[22]} = O(N)(O(C1) + O(S * C1 * N) + O(C2)) \tag{64}$$

Similarly, the complexity of reference [30] can be calculated as follows:

$$C_{[30]} = O(N)(O(D1) + O(S * D1 * N) + O(D2) + O(D3)) \tag{65}$$

In reference to [30], the complexity of Step 2 is augmented by incorporating the computation of both the coalition utility and the system utility subsequent to the inclusion of UAV $n$ to the coalitions. Meanwhile, the complexity ($D2$) of Step 3, which pertains to joining the optimal coalition, and $D3$ of Step 4, which is about updating the coalition structure, remains as small constants.

As for the proposed scheme, step 1 involves a complexity of $O(N)$ for selecting any UAV. The complexity of step 3, calculating the neighbor node's utility calculation of each action taken by the UAV $n$ as depicted in Equation (43), is $O(S * E1 * K)$ ($K < N$). Similarly, the complexity of the proposed scheme can be calculated as follows:

$$C_{proposed} = O(N)(O(E1) + O(S * E1 * K) + O(E2) + O(E3)) \tag{66}$$

Based on the above calculations, the proposed scheme boasts a lower computational complexity when compared to the references [22,30]. This can be primarily attributed to the inclusion of the local coupling effect between UAVs in the function design.

## 6. Conclusions and Future Work

In this paper, we conduct a joint investigation of UAV deployment and cache placement in a UAV-assisted network to achieve a balance between the user's satisfaction and coverage. We formulate an optimization problem that is designed to maximize the sum of the user's satisfaction and overall coverage, which we then decompose into three distinct parts. First, we use K-means to obtain initial 2D locations for the UAV and determine the optimal altitude. Second, we design a two-hop cooperative-cache mechanism to improve the MOS and employ a potential game to model the joint optimization of the user's satisfaction and coverage. To obtain optimal results, we utilize the log-linear-learning algorithm. We additionally prove that the proposed scheme is convergent. Finally, we demonstrate the feasibility and effectiveness of our proposed algorithm through simulation results, which show that the UAV deployment and cache placement improve the system performance and confirm the advantages of the two-hop cooperative cache.

The proposed method has some limitations that require attention. First, the approach outlined in this paper assumes that users have similar kinds of data requirements and employs an equal distribution strategy for bandwidth allocation, overlooking individual differences in user needs. Going forward, our research will involve addressing the challenge of managing resources for a UAV-assisted cache in the context of multi-demand user groups. Additionally, when a communication model is selected, it is assumed that various frequency bands are assigned to each UAV to prevent interference. However, a limited availability of spectrum resources necessitates the careful assessment of the impact of interference between UAVs. Furthermore, to simplify the model, we have uniformly set the height of the UAV, which will facilitate the establishment of a two-hop network for cooperative caching in the later stage. In future research, we aim to investigate cooperative caching

of UAVs at varying altitudes to provide network services. Furthermore, the endurance of UAVs and energy consumption will be a crucial factor to consider in the future.

**Author Contributions:** Conceptualization, Y.B. and J.H.; methodology and writing—original draft, Y.B.; writing—review and editing, Y.B. and S.W.; visualization Y.H. and C.F.; supervision W.L. All authors have read and agreed to the published version of the manuscript.

**Funding:** This research received no external funding.

**Data Availability Statement:** Not applicable.

**Conflicts of Interest:** The authors declare no conflict of interest.

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
