# Peer review of "Two-Hop Cooperative Caching and UAVs Deployment Based on Potential Game"

_drones, doi:10.3390/drones7070465_

Round 1

Reviewer 1 Report

This paper studied a  UAV deployment and cache placement in a UAV-assisted network to balance the user’s MOS and coverage. Initially, the optimal issue is formulated to maximize the sum of user’s MOS and the whole coverage. Afterwards, it is decomposed into 3 parts:

1.     Applying K-means to get the initial 2D location for UAV and calculate the optimal altitude.

2.     Designing the 2-hop cooperative cache mechanism to improve the MOS. Plus, applying the potential game to model the joint optimization of MOS and coverage, in which the log-linear learning algorithm is applied.

3.     Evaluating the network performance to verify the feasibility and effectiveness of the proposed algorithm by simulation results.

Pros:

·      The general idea of the manuscript is interesting.

·      The experiments are done well.

Cons:

·      The overall English must be improved.

·      The structure of the paper needs to be improved.

·      The implementation should be pointed and explained clearly.

Detailed Comments:

When defining abbreviations, the alphabets which appear in them should be in capital. For example, “Unmanned Aerial Vehicle (UAV)”, not “unmanned aerial vehicle (UAV)”. For all similar cases this point should be corrected.

The “Background and Related Works” should not be a subsection of the “Introduction” section. It needs a separate section, right after “Introduction”. Meanwhile, the bold phrases should be considered as subsections of the “Related Works” section.

In the “Related Works” section, each related work (or a few very similar ones) should be discussed in a separate paragraph. Plus, the study in this section should follow a similar approach in all paragraphs. For instance, the paragraph starts with the name of the author(s) of that related work. Now, in some paragraphs, the referencing is at the end of the last sentence, and in some others the first sentence started with the reference number.

What the authors mean by the phrase “to achieve the desirable solution” in the last paragraph of the Introduction section?

The first figure is not justified correctly in the paper.

When the authors defined an abbreviation in a sentence, in all the following parts of the paper the defined abbreviation should be used. One abbreviation should not be defined several times in the paper. A case in point is MOS. Moreover, the defined abbreviation should be appeared in the same way, in the whole paper. For example, MOS not Mos (the first line of page 20 and in figure 8).

In section 2, several equations have been defined and make reading the paper difficult for reader to follow the parameters in each. Did authors define all these equations? If not, it is better to reference some of them, instead of explaining all thoroughly.

What is “MOSE” on page 10 in the second paragraph after equation 36.

It seems that some symbols in pages 13 and 14 have not been visualized correctly, such as â–¡ and .

The last section should be “Conclusion and Future Works” and a paragraph explaining future steps should be added.

For explaining two figures in the same sentence, the authors should place an “and” between figures’ numbers. For example, “Figures 4 and 5”, not “Figure 4,5”. In all similar cases it should be corrected.

In Figure 7, the authors should set the coverage bar’s range in a way that the reader can easily follow the difference of the 3 approaches.

K-means algorithm needs to be referenced in the paper.

There are several consecutive spaces (like the first paragraph of the page 3) and free lines (mostly around equations) in the paper. They all should be removed.

The paper needs a comprehensive academic writing and English editing. Typos, like “to get the solution the problem.”, “is formulate”, and “. And”, needs to be corrected and to be edited to be clear-cut in the manuscript.

Each equation should be first pointed in the text (with a fixed format) by its number and then be appeared in the paper.

Instead of “expressed as below:” the authors should mention “expressed in the following:” or “expressed as:”.

Instead of “is set for constraint as below” the authors should mention “is set for constraint as equation (10)”.

The whole paper needs to be edited for academic English language.

Before the experimental results section, there should be a “Evolution Method” section which present the data and the criteria/metrics (in two subsections) which have been applied for evaluating the work. It is notable that they should be referenced too.

Technical Comments:

The “Related Works” section is brief. Each work should be explained more and newer references from 2023 should be added in this section.

As there are various similar clustering methods, the authors should explain why for the first step the K-means algorithm is chosen?

There should be some explanations about the Table 1. How did the authors reach out these values for the parameters? These points should be present clearly.

It is not clear how the authors implement their proposed method! There is not any technical explanation in this regard.

A link for accessing the dataset and preferably the implementation should be provided in the paper.

In the last section the authors should discuss both the strength and the limitations of the work.

The language of the study should be reconsidered comprehensively

Author Response

Dear reviewer:

Thank you for taking the time to review our manuscript and for providing us with such valuable feedback. We appreciate your insightful comments and suggestions, which will undoubtedly improve the quality of our paper. In response to your comments, we have made the following modifications to the paper, and corresponding changes are marked in yellow in the text.

For the detailed comments:

Point 1: When defining abbreviations, the alphabets which appear in them should be in capital. For example, “Unmanned Aerial Vehicle (UAV)”, not “unmanned aerial vehicle (UAV)”. For all similar cases this point should be corrected. 

Response 1: We have carefully revised abbreviations of similar cases, such as on P1 line 11   marked in yellow.

Point 2: The “Background and Related Works” should not be a subsection of the “Introduction” section. It needs a separate section, right after “Introduction”. Meanwhile, the bold phrases should be considered as subsections of the “Related Works”section.

Response 2: We have re-adjusted the structure of “Background and Related Works” part according to the Reviewer’s suggestion as shown below.

  1. Introduction

With the rapid development in 5G communication technology and the ever-increasing data demands from mobile devices, the current number and placement of ground base stations are inadequate to meet the exploding demand [1]. Particularly in challenging environments, such as those that are hostile or unknown, this not only creates economic pressure but also results in empty loads during non-off-peak traffic periods. To address this, UAVs equipped with wireless transceivers offer high mobility, deployment flexibility, and low cost, enabling them to serve as aerial base stations that can provide data services for some areas, including major events and conference activities, regardless of geography[2]. 

As a further advancement, cache-enabled UAV has become an effective solution by caching popular contents in mobile edge networks, which alleviates the access load of ground base stations, also reduces the transmission delay and improves the quality of user experience. Thus, especially in this era of data explosion, Users could fetch the requested contents from the cache-enabled UAV with less waiting time and network congestion issues. Therefore, cache-enabled UAV-assisted network is considered to be a promising technology for dealing with the relevant challenges[3,4].

  1. Related Works

In recent years, much attention has been given to cache-enabled UAV-assisted networks as a means of ensuring quality of service [5]. what’s more, there are some critical issues classified into three cases: 3D deployment, resource allocation, and UAV trajectory.

With regards to 3D deployment, there are various performance metrics, such as coverage, connectivity, energy, and throughput, which have been used as objectives for optimizing UAV deployment. For example, in [6], the authors addressed the issue of ground target coverage using UAVs while ensuring connectivity. In [7], the focus was on maximizing the coverage region of a single UAV by optimizing its vertical and horizontal dimensions and minimizing transmit power. What’s more, the work in [8] proposed the algorithm to minimize task completion time in UAV-enabled MEC systems based on SCA. In [9], potential game theory was used to control the quasi-stationary deployment of UAVs and maximize the downlink wireless coverage of a UAV swarmin an unknown mission area. Additionally, [10] considered location, power control, and connectivity to address the area coverage problem. In [11], the deployment scheme of UAVs was adjusted using Virtual Force Field theory to maximize the total network throughput based on statistical user position information.

With respect to joint 3D deployment and resource allocation, [12] addressed the joint placement of UAVs and association with users to maximize network sum-rate under bandwidth limitation and quality of service constraints. On the other hand, [13] formulated the problem of the long-term caching placement and resource allocation optimization for content delivery delay minimization as a Markov decision process solved by Q-learning. Meanwhile, in [14] the focus was on investigating jointly UAV deployment and power allocation in a UAV-assisted MIMO-NOMA WCN(wireless caching network) to minimize the user’s delay.

With regards to trajectory optimization, [15] studied the optimization of a single UAV's trajectory based on a new design paradigm of communication throughput and energy consumption. In [16], the optimization of deployment and movement for multiple UAVs was studied while considering several ground terminals (GTs) communicating with the UAVs using variable transmission power and fixed data rate. And the corresponding trajectory optimization algorithm introduced was shown to guarantee a convergent Lagrangian. [17] considered a practical 3D urban environment with imperfect CSI and designed the UAV's trajectory to minimize data collection completion time while adhering to practical throughput and flight movement constraints.

Previous studies have focused on optimizing both UAV deployment and content placement for cache placement to enhance overall network performance. In [18], an optimal configuration for content-aware UAV-assisted network content caching and location services is suggested, taking into account the correlation between users and the surroundings. This approach considers a trade-off between user’s service probability and transmission overhead when sharing cached content among one hop neighbors. However, the system model consists of a substantial central UAV and several service UAVs, with the former being accountable for delivering the cached content to designated UAVs for collaborative purposes. The research paper failed to account for the flight duration of the central UAV and the communication cost of real-time information exchange between each service UAV and central UAV.

The focus of [19] pertains to a scenario where a single UAV handles random and asynchronous content requests for ground nodes. In contrast to [18], the files were cached in specific ground nodes during the initial phase of each operation cycle and shared via device-to-device (D2D) communication. The optimization problem was aimed to minimize the weighted sum of file caching and retrieval costs by jointly designing the file caching strategy, UAV flight trajectory, and transmission scheduling. The extension of the proposed scheme to encompass multiple UAVs and the storage of files in both UAVs and ground terminals is a task was left for future exploration.

In [20], it presented a joint optimization problem aimed at maximizing the quality of experience (QoE) of users by mean opinion score (MOS), through the deployment of UAVs, caching placement, and user association. The solution was decomposed into three sub-problems, namely, swap matching based UAV deployment, greedy based caching placement, and Lagrange dual based on user association. It is worth noting that the paper provided a list of candidate UAV deployment locations.

The objective of proposed work, as outlined in reference [21], was to enhance the QoE evaluated through the content delay index (CDI) while taking into account the latency in delivering content to mobile users on the ground. To achieve this, the optimization process problem was decomposed into three stages following the procedure of optimizing the 2D position, height and proactive content caching. It could be found that UAVs serve autonomously for ground users, with no collaborative caching mechanism in place.

In their proposal, [22] proposed a three-layer cache architecture for UAVs that enabled hierarchical adaptation to the dynamic changes of users and UAVs. By utilizing a user-adaptive UAV trajectory model in the UAV based MEC layer, as well as a UAV-adaptive cache model in the cognitive center layer, both the transmission efficiency and hit rate of the system were improved. Additionally, [23] utilized mean-field game theory to optimize the placement of content in UAVs by modeling user social attributes as spatial and temporal attributes.

With regard to cache placement, previous works [20–23] have primarily focused on the two aspects: on one hand, some works investigated the impact of users’ attributes on  content placement , and utilized mathematical models to demonstrate the relationship between users and the deployment of UAVs using indicators such as transmission delay, MOS, CDI. On the other hand, other works concentrated on trajectory [24] , transmission scheduling [25] techniques with a focus on transmission power and content caching. While most of the existing research concentrated on individual UAV caching strategies or cooperative, complementary content transmission of 1-hop [26], very few studies have considered cooperative transmission among UAVs of two-hop. Such an approach could significantly reduce the number of times UAVs need to access the MBS to obtain un-cached files. Such an approach could significantly reduce the number of times UAVs need to access the MBS to obtain un-cached files. Additionally, the utility function design like in reference [20] didn’t consider the local coupling effect between the UAV and its two-hop neighborhood , and any action taken by the UAV may lead to changes in the utility values of its neighborhhod. Consequently, designing a joint, efficient, proactive 2-hop cooperative content caching and UAV deployment scheme while also taking into account the interdependence among UAVs poses a significant challenge.

Motivated by the above observations, in this paper, we propose a cache-able UAV-assisted network system where a cooperative transmission strategy and a spatial adaptive UAV deployment are jointly adopted to reduce the transmission delay and improve the coverage. Table 1 illustrates the contrasting analysis of the suggested methodology against other relevant sources. The contributions of this paper are summarized as follows:

Firstly, UAV cache placement and deployment are combinedto optimize system efficiency considering communication delay and coverage. To enhance the utility of the network, a two-hop UAV cooperative caching mechanism is proposed.

Secondly, formulate the problem of joint cooperative caching and 2D placement optimization as a strict potential game. To design the utility function of the potential game, we consider the local coupling effect resulting from action changes among UAVs. And the problem is transferred to maximize the whole network utility, which is defined by jointly considering MOS and coverage.

Reference

Cooperative Cache strategy

Cache in UAV

Metrics

Approach

Candidate location of UAVs

Notes

[18]

On-hop collaborative

User’s service and

Probability transmission overhead

Potential game

unknown

The duration of the central UAV's flight and the costs of communication for real-time information exchange between each service UAV and the central UAV have not been factored in.

[19]

×

×

Weighted sum of

file caching and retrieval costs

Swap matching

Greedy algorithm

Lagrange dual

trajectory

Only consider the single UAV which don’t provide service for users.

[20]

×

MOS

Decomposition the optimization problem

known

The coverage has not been considered.

[21]

×

CDI

Decomposition the optimization problem

unknown

The joint consideration of cache content and UAV horizontal position has not been taken into account.

[22]

×

Transmission efficiency and hit rate

Trajectory and cache model

unknown

Collaborative caching of UAVs could be considered in the future.

[23]

×

User social attributes

Mean-Field game

unknown

The mean-field game is mainly for the large number of players

[24]

×

Throughput

block alternating descent and successive convex approximation

trajectory

The joint consideration of cache content and UAV horizontal position has not been taken into account.

[25]

×

Access delay and cache-hit delay

Decomposition the optimization problem

trajectory

Collaborative cache mechanism could be considered in the future.

[26]

On-hop collaborative

transmission reliability and transmission energy consumption

Coalition Formation Game

unknown

Local coupling effect between UAVs is not considered.

proposed

two-hop collaborative

Modified MOS  and. Coverage

Decomposition the optimization problem

Potential game

unknown

Please refer to the Conclusions and Future Work section for further details.

Thirdly, the log-linear learning scheme is proposed to get the solution of the potential game.

Table 1. Comparison of literatures.

The subsequent sections of this paper are structured as follows: Section 3 presents a detailed description of the system model, while Section 4 introduces a joint proactive cooperative content caching scheme and UAV deployment strategy. The proposed approach leverages the log-linear caching algorithm to effectively achieve the desired outcomes. Simulation experiments and discussion are carried out in Section 5. And the conclusion is drawn in Section 6 finally.

Point 3: In the ‘’Related Works’’ section, each related work (or a few very similar ones) should be discussed in a separate paragraph. Plus, the study in this section should follow a similar approach in all paragraphs. For instance, the paragraph starts with the name of the author(s) of that related work. Now, in some paragraphs, the referencing is at the end of the last sentence, and in some others the first sentence started with the reference number.

Response 3As suggested by the reviewer, we have optimized the sentence structure and content of the relevant work on page 2- page 4.

Point 4What the authors mean by the phrase “to achieve the desirable solution” in the last paragraph of the Introduction section?

Response 4: We are really sorry for our inaccurate translation. And we have re-written the sentence as follows.

 The subsequent sections of this paper are structured as follows: Section 3 presents a detailed description of the system model, while Section 4 introduces a joint proactive cooperative content caching scheme and UAV deployment strategy. The proposed approach leverages the log-linear caching algorithm to effectively achieve the desired outcomes.

Point 5: The first figure is not justified correctly in the paper.

Response 5:  We are sorry for the carefulness and we have adjusted the figure on page 5.

Point 6: When the authors defined an abbreviation in a sentence, in all the following parts of the paper the defined abbreviation should be used. One abbreviation should not be defined several times in the paper. A case in point is MOS. Moreover, the defined abbreviation should be appeared in the same way, in the whole paper. For example, MOS not Mos (the first line of page 20 and in figure 8).

Response 6: We feel sorry for our carelessness. In our resubmitted manuscript, the abbreviation problem has been revised.

Point 7: In section 2, several equations have been defined and make reading the paper difficult for reader to follow the parameters in each. Did authors define all these equations? If not, it is better to reference some of them, instead of explaining all thoroughly.

Response 7: In order to make it more easily for readers to follow the parameters, we have added a glossary with definitions of the variables on page 5.

Point 8: What is  “MOSE” on page 10 in the second paragraph after equation 36.

Response 8: In our resubmitted manuscript, the typo on has been revised on page 13 line 361.

Point 9: It seems that some symbols in pages 13 and 14 have not been visualized correctly, such as â–¡ and ⊗.

Response 9: As suggested by the reviewer, we have investigated the symbols and rectified it around page 15 line 412 and page 17, line 457.

Point 10: The last section should be ”Conclusion and Future Works” and a paragraph explaining future steps should be added.

Response 10: We think this is an excellent suggestion. We have added the future steps on page 23 as follows.  

The proposed method has some limitations that require attention. Firstly, the approach outlined in this paper assumes that users have similar kind of data requirements and employs an equal distribution strategy for bandwidth allocation, overlooking individual differences in user needs. Going forward, our research will involve addressing the challenge of managing resources for UAV-assisted cache in the context of multi-demand user groups. Additionally, when selecting a communication model, it is assumed that various frequency bands are assigned to each UAV to prevent interference. However, limited availability of spectrum resources necessitates the careful assessment of the impact of interference between UAVs. Furthermore, to simplify the model, we have uniformly set the height of the UAV, which will facilitate the establishment of a two-hop network for cooperative caching in the later stage. In future research, we aim to investigate cooperative caching of UAVs at varying altitudes to provide network services. Furthermore, the endurance of UAVs and energy consumption will be a crucial factor to consider in the future.

Point 11: For explaining two figures in the same sentence, the authors should place an “and” between figures’ numbers. For example, “Figures 4 and 5”, not “Figure 4,5” . In all similar cases it should be corrected.

Response 11: We are really sorry for our inaccurate way of writing. And we have re-written the sentence in our resubmitted manuscript such as the example on page 20, line 508-509.

Point 12: In Figure 7, the authors should set the coverage bar’ s range in a way that the reader can easily follow the difference of the 3 approaches.

Response 12: As suggested by the reviewer, we have revised the figure for better readability and understanding on page 22, Figure 7.

Point 13: K-means algorithm needs to be referenced in the paper.

Response 13: We have referenced the K-means algorithm in our resubmitted manuscript in page 11, line 316-line 321 as follows.

To guarantee the connection probability between UE and UAV, the ground users are clustered into several subsets using K-means [37]clustering algorithm. The process involves initializing k cluster centers, calculating the Euclidean distance between each user and the cluster center using formula (31), assigning each user to the closest cluster center. Therefore, the initial horizontal coordinates of UAVs are then determined as the center of each user cluster.

(31)

Point 14: There are several consecutive spaces (like the first paragraph of the page 3) and free lines (mostly around equations) in the paper. They all should be removed.

Response 14: We feel sorry for our carelessness. In our resubmitted manuscript, all consecutive spaces and free lines have be removed from the paper.

Point 15: The paper needs a comprehensive academic writing and English editing. Typos, like “to get the solution the problem.”, “is formulate”, and “. And”, needs to be corrected and to be edited to be clear-cut in the manuscript.

Point 16: The whole paper needs to be edited for academic English language.

Response 15 and 16: We have carefully revised and polished the language of our paper and have made the necessary changes for the language errors, such as the errors you pointed out on page 16 line 416-line 417,and page 23, line 563 as follows.

The log-linear based learning rule is a classic scheme designed to achieve the optimal policy solution. 

We formulate an optimization problem that is designed to maximize the sum of the user's MOS and overall coverage, which we then decompose into three distinct parts.

Point 17: Each equation should be first pointed in the text (with a fixed format) by its number and then be appeared in the paper.

Response 17: We have carefully revised the format of the paper in page 16, Equation(51).

Point 18: Instead of “expressed as below:” the authors should mention “expressed in the following:” or “expressed as:”.

Response 18: We feel sorry for our inaccurate English grammar, all similar expressions have be revised and marked in yellow in the resubmitted manuscript,such as the errors you pointed out on page 6, line 210.

A common approach to modeling the air-to-ground propagation channel involves analyzing the LOS and NLOS components separately and considering their respective probabilities of occurrence, which are denoted by:

Point 19: Instead of “is set for constraint as below” the authors should mention “is set for constraint as equation (10)”.

Response 19: This sentence has been amended in the paper P7, line 224 -line 225.

 The total number of users associated with UAV n is denoted by . To account for the link quality between UAV n and ground user UE i, a threshold  is set for constraint as Equation (10).

Point 20: Before the experimental results section, there should be a “Evolution Method” section which present the data and the criteria/metrics (in two subsections) which have been applied for evaluating the work. It is notable that they should be referenced too.

Response 20:

We have carefully cited the source of some parameters of the experiment, and described the strategy of comparison in detail on page 18, line 463-line 478.

For the detailed comments:

Technique comment 1: Please refer to the response 2 and 3 above.

Technique comment 2: Please refer to the response 13 above.

Technique comment 3: Please refer to the response 20 above.

Technique comment 4: We have clarify the scheme in the page 16 in the resubmitted paper.

Technique comment 5: Please refer to the response 20 above.

Technique comment 6: We have compared the proposed scheme with similar papers on page 23 in the resubmitted paper

Reviewer 2 Report

This paper explores the joint cache placement and 3D deployment of clusters of unmanned aerial vehicles (UAVs) using potential game theory, considering the cooperative caching mechanism of two-hop UAVs, which has the potential to reduce latency and enhance the energy efficiency of transmission at certain hot spots.

Although the article is well presented with a good idea, it remains unclear in several aspects. First of all, the abstract should contain the results of your simulations.

- your introduction needs improvement regarding the way you study literature articles. it is important to illustrate the disadvantages of some articles compared to yours in a table.

-In your motivation, you used references [20] [23] [24] [25] [26], what is your novelty compared to these articles.

-The system model you are proposing is not clear. it is important to formulate your model (network model) well. is it that the users communicate only with the drones. if this is the case, i find that in reality it is too complicated to ensure communication for a long time. indeed, the drones are limited by their battery and even with energy harvesting, they can not ensure continuous communication. 

-The model you propose sets the height of all drones. however, in some regions, drones must be at their maximum height to ensure LoS links. i find that setting the height is not realistic. see '' Downlink Performance Analysis in MIMO UAV-Cellular Communication With LOS/NLOS Propagation Under 3D Beamforming," 

-Explain your choice of propagation model.

-  you Assume that the available bandwidth ?0 of UAVs is divided equally among users. why such an approach, given that the service may be different. a user watching video needs more bandwidth than a user using voice.

- your model does not take into account interference between UAVs or interference between UAVs and macro base station, which makes your model too simplified.

- I find that the theory you apply has already been used in many articles. What is your mathematical novelty in the theorem?

-"the deviation of for" please correct "the derivation" your English needs revision.

- Equation 33 is not clear, it is important to say why you are making this derivation and explain equation 33.

- The convergence of your algorithm should be compared to other approaches. The same goes for your results, which must be compared with other results.

- The article needs to be organised in a way that is easy to understand 

- reference [2] and [3] are the same. please correct 

please cite this paper

-Modeling and Analysis of UAV-Assisted Mobile Network with Imperfect Beam Alignment

Interference modeling for low-height air-to-ground channels in live LTE networks

Performance analysis of UAV multiple antenna-assisted small cell network with clustered users.

Measurement-Based Propagation Channel Modeling for Communication between a UAV and a USV

nterference modeling for low-height air-to-

Extensive english are  required 

Author Response

Dear reviewer:

Thank you for taking the time to review our manuscript and for providing us with such valuable feedback. We appreciate your insightful comments and suggestions, which will undoubtedly improve the quality of our paper. In response to your comments, we have made the following modifications to the paper, and corresponding changes are marked in red in the text.

Point 1: your introduction needs improvement regarding the way you study literature articles. it is important to illustrate the disadvantages of some articles compared to yours in a table.

Response 1: Thanks for your advice, we have added the literature comparison in our resubmitted paper on Page 2 as follows.

 Table 1. Comparison of literatures.

Reference

Cooperative Cache strategy

Cache in UAV

Metrics

Approach

Candidate location of UAVs

Notes

[18]

On-hop collaborative

User’s service and

Probability transmission overhead

Potential game

unknown

The duration of the central UAV's flight and the costs of communication for real-time information exchange between each service UAV and the central UAV have not been factored in.

[19]

×

×

Weighted sum of

file caching and retrieval costs

Swap matching

Greedy algorithm

Lagrange dual

trajectory

Only consider the single UAV which don’t provide service for users.

[20]

×

MOS

Decomposition the optimization problem

known

The coverage has not been considered.

[21]

×

CDI

Decomposition the optimization problem

unknown

The joint consideration of cache content and UAV horizontal position has not been taken into account.

[22]

×

Transmission efficiency and hit rate

Trajectory and cache model

unknown

Collaborative caching of UAVs could be considered in the future.

[23]

×

User social attributes

Mean-Field game

unknown

The mean-field game is mainly for the large number of players

[24]

×

Throughput

block alternating descent and successive convex approximation

trajectory

The joint consideration of cache content and UAV horizontal position has not been taken into account.

[25]

×

Access delay and cache-hit delay

Decomposition the optimization problem

trajectory

Collaborative cache mechanism could be considered in the future.

[26]

On-hop collaborative

transmission reliability and transmission energy consumption 

Coalition Formation Game

unknown

Local coupling effect between UAVs is not considered.

proposed

two-hop collaborative

Modified MOS  and. Coverage

Decomposition the optimization problem

Potential game

unknown

Please refer to the Conclusions and Future Work section for further details.

Point 2: In your motivation, you used references [20] [23] [24] [25] [26], what is your novelty compared to these articles.

Response 2: Contrasting with the approaches taken in references [18] and [26], which created a one-hop cache cooperative mechanism, or [21]-[25], which operated UAVs independently to provide services, we propose an service strategy using UAV two-hop neighborhood caching in our design of cooperative caching mechanism for UAVs. This approach effectively reduces transmission pressure on the backhaul transmission network. Additionally, unlike the utility function design in reference [20], we have devised a local coupling mechanism between the UAV and its two-hop neighborhood to inform the UAV's utility function. As such, any action taken by the UAV may lead to changes in the utility values of its two-hop nodes. And We have already rewritten the abstract and conclusion parts, as shown as below:

Abstract

This paper explores the joint cache placement and 3D deployment of Unmanned Aerial Vehicle (UAV) groups, utilizing potential game theory and a 2-hop UAV cooperative caching mechanism. which could makes tradeoff between latency and coverage. The proposed scheme consists of three parts: firstly, the initial 2D location of UAV groups is determined through K-means, with the optimal altitude determined based on the UAV coverage radius.Secondly, in order to balance the transmission delay and coverage, the MOS (Mean Opinion Score) and coverage are designed to evaluate the performance of UAV-assisted networks. Then, the potential game is modeled which transfers the optimization problem into the maximization of the whole network utility. The local coupling effect resulting from action changes among UAVs is considered in the design of the potential game utility function of the. Moreover, the log-linear learning scheme is applied to solve the problem. Finally, the simulation results verify the superiority of the proposed scheme, in terms of the achievable transmission delay and coverage performance compared with other two tested schemes. The coverage ratio is close to 100% when the UAV number is 25, and the User number is 150, in addition, it outperforms the benchmarks when it comes to maximizing MOS of users.

Conclusions and Future Work

The proposed method has some limitations that require attention. Firstly, the approach outlined in this paper assumes that users have similar kind of data requirements and employs an equal distribution strategy for bandwidth allocation, overlooking individual differences in user needs. Going forward, our research will involve addressing the challenge of managing resources for UAV-assisted cache in the context of multi-demand user groups. Additionally, when selecting a communication model, it is assumed that various frequency bands are assigned to each UAV to prevent interference. However, limited availability of spectrum resources necessitates the careful assessment of the impact of interference between UAVs. Furthermore, to simplify the model, we have uniformly set the height of the UAV, which will facilitate the establishment of a two-hop network for cooperative caching in the later stage. In future research, we aim to investigate cooperative caching of UAVs at varying altitudes to provide network services. Furthermore, the endurance of UAVs and energy consumption will be a crucial factor to consider in the future.

Point 3: -The system model you are proposing is not clear. it is important to formulate your model (network model) well. is it that the users communicate only with the drones. if this is the case, i find that in reality it is too complicated to ensure communication for a long time. indeed, the drones are limited by their battery and even with energy harvesting, they can not ensure continuous communication.

Response 3: We have made clearly description of the system model in our resubmitted manuscript on line 169-line 174. In considering the endurance of drones, energy consumption will be regarded as a pivotal factor in the future, as highlighted in the conclusion and future work section on page 24, line 573- line 586.

Point 4: -The model you propose sets the height of all drones. however, in some regions, drones must be at their maximum height to ensure LoS links. i find that setting the height is not realistic. see '' Downlink Performance Analysis in MIMO UAV-Cellular Communication With LOS/NLOS Propagation Under 3D Beamforming,"

Response 4: We appreciate you bringing the research paper titled "Downlink Performance Analysis in MIMO UAV-Cellular Communication With LOS/NLOS Propagation Under 3D Beamforming" to my attention. I have thoroughly reviewed the contents and found it to be informative and relevant to our current project. Thank you for sharing it with me. In fact, to simplify the model, we have uniformly set the height of the UAV, which will facilitate the establishment of a two-hop network for cooperative caching in the later stage. In future research, we plan to investigate cooperative caching of UAVs at varying altitudes with the aim of providing network services.

Point 5: -Explain your choice of propagation model.

Response 5: We have made a explanation in the resubmitted paper on page 6, line 199- line 210; page 7, line 227- line234; page 8, 243-248.

Point 6: - you Assume that the available bandwidth B0 of UAVs is divided equally among users. why such an approach, given that the service may be different. a user watching video needs more bandwidth than a user using voice.

Response 6: Thank you for bringing up a valid point. In our paper, we did not consider the varying needs of different users and instead assumed that all users require the same type of service. We simply allocated bandwidth equally among users for the sake of simplicity, but this may not accurately reflect real-life scenarios. Our future work will take into account the diverse sets of user needs.

Point 7: - your model does not take into account interference between UAVs or interference between UAVs and macro base station, which makes your model too simplified.

Response 7: Regarding the issue of interference among UAVs, we have assumed that assigning different frequencies to different UAVs can help avoid annoyance and interference which hasn’t been stated in the paper. Therefor, we add the specification of the above assumption on page 8, line 235 - line239. Actually, considering interference is highly valuable for practical applications, particularly as spectrum resources are limited. Therefore, in our future work, we will also take into account interference between UAVs.

Point 8: - I find that the theory you apply has already been used in many articles. What is your mathematical novelty in the theorem?

Response 8: On one hand, we have implemented a new UAV caching strategy in which the UAV leverages a two-hop cooperative caching mechanism to provide better services to users while reducing the pressure on the MBS to UAV link. On the other hand, we have explored the local coupling mechanism between the UAV and its two-hop neighborhood in designing the UAV's utility function as expressed in Equation (41), where any action taken by the UAV can result in a change in its two-hop nodes’ utility value.

(41)

Point 9 and 10: -"the deviation of for" please correct "the derivation" your English needs revision.

-The article needs to be organised in a way that is easy to understand.

Response 9 and 10: Firstly, we have carefully revised and polished the language of our paper and have made the necessary changes for the language errors, such as the errors you pointed out on page 11, line 331-line 332. Additionally, we have restructured the “Background and Related Works” section to provide a more coherent flow of ideas. To elucidate the parameters of Section 3, we have appended a glossary containing definitions of the variables. Lastly, we have revised the final section as “Conclusion and Future Work”.

Point 11: - Equation 33 is not clear, it is important to say why you are making this derivation and explain equation 33.

Response 11: Equation (33)is shown as follows, indicating that the path loss between the UAV and the user has a maximum value .In fact, the maximum path loss can be determined from the above equation as follows. 

(33)

In order to determine the optimal altitude point for achieving the best coverage, h can be expressed in the following manner.

(34)

(35)

Then, compute the first and second partial derivatives of h with respect to , denoted as follows respectively:

(36)

In this case, the optimal altitude of UAV is approximated with the determined values of UAV coverage radius and . 

(37)

(38)

Point 12: - The convergence of your algorithm should be compared to other approaches. The same goes for your results, which must be compared with other results.

Response 12: We have made a detailed comparison of the similar schemes in our resubmitted paper on page 22-23, line 552-587.

Point 13: - reference [2] and [3] are the same. please correct. 

Response 13: We feel sorry for our carelessness. In our resubmitted manuscript, the reference has been corrected.  

  • Jiang X , Sheng M , Zhao N , et al. Green UAV communications for 6G: A survey. Chinese Journal of Aeronautics 2022, 35(9), 16.

Point 14: please cite this paper

-Modeling and Analysis of UAV-Assisted Mobile Network with Imperfect Beam Alignment

-Interference modeling for low-height air-to-ground channels in live LTE networks

-Performance analysis of UAV multiple antenna-assisted small cell network with clustered users.

-Measurement-Based Propagation Channel Modeling for Communication between a UAV and a USV

Response 14: As suggested by the reviewer, we have added more references to support this idea.

 [32] M. A. Ouamri, R. Alkanhel, C. Gueguen, M. A. Alohali and S. S. M. Ghoneim. Modeling and analysis of uav-assisted mobile network with imperfect beam alignment. Computers, Materials & Continua 2023, 74(1), 453–467.

[34]X. Cai, C. Zhang, J. Rodríguez-Piñeiro, X. Yin, W. Fan and G. F. Pedersen. Interference Modeling for Low-Height Air-to-Ground Channels in Live LTE Networks. IEEE Antennas and Wireless Propagation Letters 2019, 18(10), 2011-2015.

  • Ouamri, M.A., Singh, D., Muthanna, M.A. et al. Performance analysis of UAV multiple antenna-assisted small cell network with clustered users. Wireless Network2023, 29, 1859–

[29]Y. Yu, J. Rodríguez-Piñeiro, X. Shunqin, Y. Tong, J. Zhang and X. Yin. Measurement-Based Propagation Channel Modeling for Communication between a UAV and a USV. In Proceedings of 2022 16th European Conference on Antennas and Propagation (EuCAP), Madrid, Spain, 27 March - 01 April 2022; pp. 01-05,

Reviewer 3 Report

This paper gives a substantial investigation of the performance of the 2-hop cooperative caching and UAVs deployment based on the potential game. The authors try to make a tradeoff between latency and coverage. The theoretical derivation is rigorous and the results appear to be correct and believable. In general, the manuscript (it seems a revised manuscript marked with highlighted color) provides an interesting conclusion. Nevertheless, the reviewer holds some concerns about this work which you can find below. I suggest that the authors revise and improve the manuscript accordingly.

In my opinion, the channel model or channel state information of the transmission link is significant for channel estimation and even for UAV trajectory design. However, this is ignored in the Introduction section. The following references could help the readers with previously undertaken research works, such as doi: 10.1109/TVT.2023.3252822, 10.1109/JLT.2013.2293137.

There are certain typographical and grammatical errors throughout the manuscript. The authors are suggested to proofread the paper. In addition, the authors are advised to recheck the manuscript once and remove those errors.

For example, 1-hop--> single-hop, 2-hop or two-hop -->dual-hop. 

The proposed scheme in the legend of each figures --> Proposed scheme.

“… utilizing potential game theory and a 2-hop UAV cooperative caching mechanism. which …" There should be a comma before which.

“… [5]. what’s more, …” ---> “What’s more,”

“Then, compute the first and second partial derivatives of h with respect to ?, denoted as follows respectively:” Please clarify this sentence.

The mathematical writing needs to be improved. Below is just an incomplete list of suggestions:

(a) It would be good to use romantic numbers for lists.

(b) It would be necessary to number each equation and not start a sentence with a math symbol.

(c) It would be better to use capital letters for random variables and little letters for realizations.

(d) It would be necessary to use parenthesis or brackets large enough and align equations with equal signs.

(e) It would be necessary to use mathrm for non-math content in equations.

Minor editing of English language required

Author Response

Dear reviewer:

Thank you for taking the time to review our manuscript and for providing us with such valuable feedback. We appreciate your insightful comments and suggestions, which will undoubtedly improve the quality of our paper. In response to your comments, we have made the following modifications to the paper, and corresponding changes are marked in yellow in the text.

Point 1: In my opinion, the channel model or channel state information of the transmission link is significant for channel estimation and even for UAV trajectory design. However, this is ignored in the Introduction section. The following references could help the readers with previously undertaken research works, such as doi: 10.1109/TVT.2023.3252822, 10.1109/JLT.2013.2293137.

Response 1: As suggested by the reviewer, we have added more references and part of the literature on channel modeling and channel performance metrics as follows.

In recent years, significant attention has been focused on cache-enabled UAV-assisted networks as a means of ensuring quality of service [5]. Extensive research has been conducted on various aspects, including the channel state information and performance, such as the communication model, bit error rate (BER), and channel capacity. In [6], a statistical propagation model was proposed to predict the air-to-ground path loss between a low altitude platform and a terrestrial terminal, which characterized the air-to-ground path into two distinct path loss profiles. Furthermore, [7] provided statistical models for air-to-ground radio channels in dense urban environments, demonstrating that airborne platforms can act as relaying nodes to extend the range and improve connectivity between terrestrial ad-hoc terminals. Additionally, [8,9] presented an elaborate analysis of mixed RF/FSO systems, providing an integrated investigation of UAV-assisted wireless communication systems. 

[8] G.Xu, N. Zhang, M. Xu, Z. Xu, Q. Zhang and Z. Song. Outage Probability and Average BER of UAV-assisted Dual-hop FSO Communication with Amplify-and-Forward Relaying. IEEE Transactions on Vehicular Technology 2023, 1-16.

[9] T. Song and P. -Y. Kam. A Robust GLRT Receiver With Implicit Channel Estimation and Automatic Threshold Adjustment for the Free Space Optical Channel with IM/DD. Journal of Lightwave Technology 2014, 32(3), 369-383.

Point 2: There are certain typographical and grammatical errors throughout the manuscript. The authors are suggested to proofread the paper. In addition, the authors are advised to recheck the manuscript once and remove those errors.

(a)For example, 1-hop--> single-hop, 2-hop or two-hop -->dual-hop.

(b)The proposed scheme in the legend of each figures --> Proposed scheme.

(c)“… utilizing potential game theory and a 2-hop UAV cooperative caching mechanism. which …" There should be a comma before which.

(d)“… [5]. what’s more, …’ ---> “What’s more,”

Response 2: We have carefully revised and polished the language of our paper and have made the necessary changes for the language errors.

(a)We have revised abbreviations used in similar cases, such as the one onpage 1, line 12, which has been highlighted in yellow.

(b) We have revised the legend of each figures, from page 20 to page 23.

(c) We have added the comma in line 13 to ensure the sentence is complete and has correct grammar, which has been highlighted in yellow.

(d) The initial letter that was not capitalized has been rectified on page 2, line 58.

Point 3:”Then, compute the first and second partial derivatives of h with respect to , denoted as follows respectively:” Please clarify this sentence.

Response 3: We feel sorry for our carelessness. In our resubmitted manuscript, the sentence has been corrected on page 12, line 340.

Then, compute the first order partial derivatives of h with respect to  , denoted as follows respectively:

Point 4: The mathematical writing needs to be improved. Below is just an incomplete list of suggestions:

(a) It would be good to use romantic numbers for lists.

(b) It would be necessary to number each equation and not start a sentence with a math symbol.

(c) It would be better to use capital letters for random variables and little letters for realizations.

(d) It would be necessary to use parenthesis or brackets large enough and align equations with equal signs.

(e) It would be necessary to use mathrm for non-math content in equations.

Response 4:

(a)As suggested by the reviewer, we have used romantic numbers to list the different types of communication link in line 208, line 234, line 250.  

(b) We have carefully revised and modified the sentences to introduce the equation or mathematical expression in a complete and grammatically correct sentence before using any symbols, such as the one on page 7, line 231.

(c) Thanks for your suggestion. We have thoroughly examined the random variables and realizations, and have made the necessary revisions, such as the ones on page 16, line 434 and page 17, Equation (56)-(58).

(d) As suggested by the reviewer, we have adjusted the formula format to get it aligned in page 11, Equation (32).

(e) We have deleted the non-math content on page 17, Equation (56).

Round 2

Reviewer 2 Report

The paper have been improved 

minors revision 

Author Response

Dear reviewer:

Thank you for taking the time to review our manuscript and for providing us with such valuable feedback. We appreciate your insightful comments and suggestions, which will undoubtedly improve the quality of our paper. In response to your comments, we have made the following modifications to the paper, and corresponding changes are marked in yellow in the text.

Point : Minor editing of English language required

Response : We have meticulously reviewed and refined the editing of the English language, including the one found on page 9, lines 272-275, as well as page 19, lines 500-509.
